# Targeting ferroptosis by poly(acrylic) acid coated Mn$_3$O$_4$ nanoparticles alleviates acute liver injury

Xinyi Shan[1,2,7], Jiahuan Li[1,2,7], Jiahao Liu[2,3,4,7], Baoli Feng[1,2], Ting Zhang[1,2], Qian Liu[1,2], Huixin Ma[2,3], Honghong Wu [2,3,5,6] ✉ & Hao Wu [1,2] ✉

Ferroptosis, a newly characterized form of regulated cell death, is induced by excessive accumulation of lipid peroxidation catalyzed by intracellular bioactive iron. Increasing evidence has suggested that ferroptosis is involved in the pathogenesis of several human diseases, including acute liver injury. Targeted inhibition of ferroptosis holds great promise for the clinical treatment of these diseases. Herein, we report a simple and one-pot synthesis of ultrasmall poly(acrylic) acid coated Mn$_3$O$_4$ nanoparticles (PAA@Mn$_3$O$_4$-NPs, PMO), which perform multiple antioxidant enzyme-mimicking activities and can scavenge broad-spectrum reactive oxygen species. PMO could potently suppress ferroptosis. Mechanistically, after being absorbed mainly through macropinocytosis, PMO are largely enriched in lysosomes, where PMO detoxify ROS, inhibit ferritinophagy-mediated iron mobilization and preserve mTOR activation, which collectively confer the prominent inhibition of ferroptosis. Additionally, PMO injection potently counteracts lipid peroxidation and alleviates acetaminophen- and ischaemia/reperfusion-induced acute liver injury in mice. Collectively, our results reveal that biocompatible PMO act as potent ferroptosis inhibitors through multifaceted mechanisms, which ensures that PMO have great translational potential for the clinical treatment of ferroptosis-related acute liver injury.

Ferroptosis is a newly characterized form of nonapoptotic programmed cell death that was discovered in the last decade[1]. This process is distinct from other well-defined types of regulated cell death at the morphological, genetic and biochemical levels. Lipid peroxidation, the fundamental feature of ferroptosis, is nonenzymatically catalyzed by intracellular bioactive iron or enzymatically catalyzed by LOXs (lipoxygenases)[2] and POR (cytochrome P450 oxidoreductase)[3,4].

The lethal accumulation of lipid peroxides substantially disrupts the ion gradients across the biomembrane, decreases membrane fluidity, and increases membrane permeability to initiate the death program, although the detailed mechanism is only partially understood[5]. Conversely, the cellular antioxidant systems, especially the GPX4-GSH[6], FSP1-CoQH2 (ubiquinol)-NAD(P)H[7,8], GCH1-tetrahydrobiopterin[9], and DHODH-ubiquinone[10] axes, constitute the major defense mechanisms

[1]State Key Laboratory of Agricultural Microbiology, College of Veterinary Medicine, Huazhong Agricultural University, Wuhan 430070, China. [2]Hubei Hongshan Laboratory, Wuhan 430070, China. [3]MOA Key Laboratory of Crop Ecophysiology and Farming System in the Middle Reaches of the Yangtze River, College of Plant Science & Technology, Huazhong Agricultural University, Wuhan 430070, China. [4]College of Agriculture, Tarim University, Alar 843300, China. [5]Shenzhen Institute of Nutrition and Health, Huazhong Agricultural University, Wuhan 430070, China. [6]Shenzhen Branch, Guangdong Laboratory for Lingnan Modern Agriculture, Genome Analysis Laboratory of the Ministry of Agriculture, Agricultural Genomics Institute at Shenzhen, Chinese Academy of Agricultural Sciences, Shenzhen 518120, China. [7]These authors contributed equally: Xinyi Shan, Jiahuan Li, Jiahao Liu.
✉e-mail: honghong.wu@mail.hzau.edu.cn; whao.1988@mail.hzau.edu.cn

to trap lipophilic radicals and counteract ferroptosis. To date, compelling evidence has emerged that excessive ferroptosis is implicated in the pathogenesis of many human diseases. Therefore, pharmacological inhibition of ferroptosis holds great promise for the clinical treatment of these diseases[5].

Vulnerability to ferroptosis is mediated by several metabolic pathways, especially cellular iron metabolism[5]. Lipid peroxidation is largely determined by intracellular labile iron, which is coordinated by iron uptake, iron metabolism (mainly referring to the biosynthesis of heme and iron-sulfur clusters), iron mobilization and iron efflux. Iron overload is a typical hallmark of ferroptosis, and the concept of ferroptosis was originally coined due to the observation that this cell death could be abolished by iron chelation[1]. Intracellular bioactive iron converts $H_2O_2$ to the more reactive hydroxyl radical (OH•), which can attack PUFA (polyunsaturated fatty acid)-containing phospholipids and catalyze the generation of lipid peroxides. Additionally, iron can sustain the enzymatic activities of LOXs and POR, indirectly facilitating lipid peroxidation. A large portion of intracellular iron is sequestered by the ferritin nanocages in the cytosol. It has been reported that ferritin-bound iron can be mobilized through selective autophagy to degrade ferritin (which is known as ferritinophagy). Specifically, when intracellular iron is deficient, the ferritinophagy receptor NCOA4 (nuclear receptor coactivator 4) is stabilized, interacts with ferritin and delivers ferritin to be engulfed by the autophagosomes for lysosomal degradation and iron release[11,12]. Our study[13] and others[14,15] reported that ferritinophagy was triggered to complement cellular bioactive iron during ferroptosis induction. Genetic ablation of key autophagy components or the specific ferritinophagy receptor NCOA4 significantly reduced the labile iron pool and desensitized cells to ferroptosis, indicating that intracellular iron recycling is fundamentally important for ferroptosis.

Increasing evidence has suggested that excessive ferroptosis is implicated in the pathogenesis of many human diseases. Acute liver failure, which is a rare but life-threatening illness, is generally caused by drug overdose, viral (hepatitis A, B, and E viruses) infections, alcohol abuse and other factors[16]. It was recently reported that excessive exposure to APAP (acetaminophen, a widely used analgesic and antipyretic medication) induces acute hepatotoxicity, during which the major ferroptosis characteristics, including lipid peroxidation, GSH exhaustion and GPX4 inactivation, are typically observed in the liver. Importantly, the iron chelator deferoxamine and the specific ferroptosis inhibitor Fer-1 (ferrostatin-1) could eliminate these ferroptosis characteristics and alleviate acute hepatotoxicity[17,18]. Additionally, a similar phenomenon has been observed in the liver during ischemia/reperfusion-induced damage[19,20]. These studies collectively suggest that scavenging lipid peroxides and suppressing ferroptosis would be clinically beneficial for the treatment of acute liver injury. However, due to their low biostability, biosafety and solubility, current ferroptosis inhibitors are not suitable for clinical application[21].

Due to the high catalytic ability, facile modification and low manufacturing cost, some biocompatible nanomaterials have been applied in biomedical fields[22]. It has been widely reported that certain nanomaterials perform one or several enzymatic activities of CAT (catalase), SOD (superoxide dismutase) and GPX (glutathione peroxidase), and can scavenge the corresponding types of ROS (reactive oxygen species) to treat oxidative damage-related diseases[23]. As an essential trace element, Mn (manganese) participates in various physiological processes. In particular, Mn serves as a cofactor of SOD and plays an important role in maintaining intracellular redox homeostasis[24]. In the last decade, Mn-based nanomaterials have shown versatile applications in biomedical fields due to their good biocompatibility and high ROS-scavenging properties[25,26]. However, whether Mn-based nanomaterials can inhibit ferroptosis and be used to treat ferroptosis-related diseases has not been investigated.

In this study, we report the one-pot synthesis of biocompatible poly(acrylic) acid coated $Mn_3O_4$ nanoparticles (PAA@$Mn_3O_4$-NPs, PMO). PMO could scavenge broad-spectrum ROS, especially OH•, thereby efficiently reducing lipid peroxidation and counteracting ferroptotic cell death. Mechanistically, after being absorbed mainly through macropinocytosis, PMO largely resided in lysosomes, where PMO scavenged ROS, inhibited ferritinophagy-mediated iron mobilization, and preserved mTOR phosphorylation. Furthermore, intravenously injected PMO showed good biocompatibility and were mainly enriched in the liver. More importantly, PMO substantially inhibited hepatic ferroptosis and alleviated APAP-induced and ischemia/reperfusion-induced acute liver injury. Overall, the good biocompatibility, easy production and robust inhibition of ferroptosis by PMO make them promising candidates for the treatment of ferroptosis-associated diseases, especially APAP-induced liver injury and liver ischemia/reperfusion injury.

## Results

### Synthesis and characterization of PMO

Figure 1a shows a schematic diagram of the one-pot synthesis of the $Mn_3O_4$ nanoparticles. Briefly, $MnSO_4·H_2O$ and poly(acrylic) acid were mixed and autoclaved for one-pot synthesis. The obtained poly(acrylic) acid coated $Mn_3O_4$ nanoparticles (PAA@$Mn_3O_4$-NPs, PMO) were purified and further characterized. TEM (transmission electron microscopy) imaging showed that PMO were spherical and exhibited good dispersibility, with a TEM size of $6.1 ± 2.2$ nm (Fig. 1b). High-resolution TEM imaging further showed that the d space of PMO was 0.28 nm (inset of Fig. 1b). To further characterize the size and surface charge of PMO in the medium, the hydrodynamic diameter and zeta potential of PMO were measured with a Brookhaven Zeta-sizer. Our results showed that the hydrodynamic diameter and zeta potential of PMO were $10.2 ± 1.8$ nm (polydispersity index: $0.14 ± 0.01$) and $-38.3 ± 0.2$ mV, respectively (Fig. 1c, d). The FTIR (Fourier transform infrared spectroscopy) results confirmed the existence of -OH and C=O groups in PMO, suggesting that PAA was successfully coated on the $Mn_3O_4$ nanoparticles (Supplementary Fig. 1).

To evaluate the ROS scavenging ability of PMO, in vitro ROS scavenging assays were performed. As shown in Fig. 1e–g, PMO exhibited CAT-like $H_2O_2$ scavenging, SOD-like $O_2^{·-}$ scavenging, and OH• scavenging (OH• is not directly detoxified by any known antioxidant enzymes in biological systems) activities in a dose-dependent manner. At a concentration of 50 µg/mL, PMO scavenged 15.12% $H_2O_2$, 46.84% $O_2^{·-}$, and 57.33% OH• (Fig. 1e–g). These results demonstrated that PMO performed antioxidant enzyme-mimicking activities to scavenge broad-spectrum ROS, especially OH•.

### PMO effectively counteracts ferroptosis

As mentioned previously, ferroptosis is a newly characterized form of regulated cell death. Oxidative insult and intracellular bioactive iron facilitate the generation of lipid peroxides to induce the death program. Scavenging ROS by activating endogenous antioxidant systems or by supplementing exogenous antioxidants can trap lipid peroxides and inhibit ferroptosis[5]. Because PMO perform multiple antioxidant enzyme-mimicking activities to detoxify broad-spectrum ROS, we evaluated the potential of PMO to inhibit ferroptosis. First, PMO supplementation had no obvious cytotoxicity on MEFs (mouse embryonic fibroblasts), even at a higher concentration (200 µg/mL), as shown by microscopy imaging and cell viability assays (Supplementary Fig. 2a), indicating the good cytocompatibility of PMO. RSL3 is a typical ferroptosis inducer that covalently binds to and inhibits GPX4[6]. RSL3 exposure induced cell death in MEFs, as shown by elevated PI (propidium iodide)-positive staining and decreased cell viability. This cell death was almost completely suppressed by Fer-1, a specific ferroptosis inhibitor (Fig. 2a, b), suggesting that this type of cell death was indeed ferroptosis. Importantly, PMO supplementation (50 µg/mL)

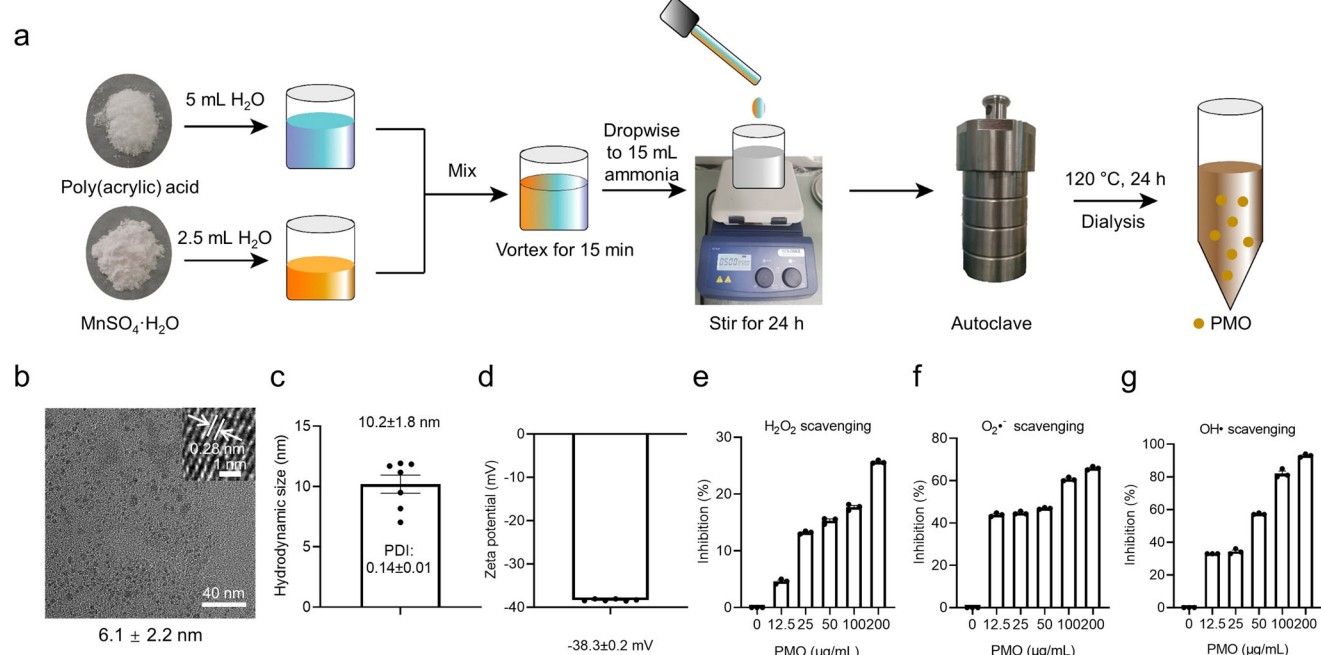

**Fig. 1 | Synthesis and characterization of PMO. a** Schematic illustration of PMO synthesis. **b** TEM image of PMO. The inserted sub-image is the d space of PMO. The TEM imaging was repeated three times independently with similar results. **c, d** Hydrodynamic size and zeta potential of PMO. **e–g** $H_2O_2$, $O_2^{\cdot-}$ and OH• scavenging activity of PMO. For statistical analysis, data represent mean ± SEM in **c–g**. $n = 7$ independent experiments in **c**, $n = 6$ independent experiments in **d**, and $n = 3$ samples in **e–g**. Source data are provided as a Source Data file.

effectively counteracted RSL3-induced ferroptosis (Fig. 2a, b). Erastin is another widely used ferroptosis inducer. It triggers ferroptosis by inhibiting the cystine/glutamate antiporter System Xc⁻, leading to limited cystine uptake and reduced GSH biosynthesis[1]. PMO supplementation (50 µg/mL) prominently suppressed erastin-induced ferroptosis (Fig. 2c, d). Furthermore, cell death analysis as evidenced by PI staining and cell viability analysis showed that PMO inhibited RSL3- and erastin-induced ferroptosis in a dose-dependent manner (Supplementary Fig. 2b, c). PMO supplementation led to long-time protection. RSL3 and erastin killed most cells in 6 h and 8 h, respectively, while the ferroptosis was even suppressed after 16 h of exposure to RSL3 or erastin in the presence of PMO (Supplementary Fig. 2d, e). The human fibrosarcoma cell line HT1080 is highly sensitive to ferroptosis and widely used in the field of ferroptosis study. We also found that PMO supplementation protected HT1080 cells against RSL3- and erastin-induced ferroptosis (Supplementary Fig. 2f, g).

ROS burst and the excessive accumulation of lipid peroxides are typical characteristics of ferroptosis. By using BODIPY 581/591-C11, a fluorescent reporter of lipid peroxidation, we found that RSL3 and erastin increased lipid peroxidation, and this effect could be largely alleviated by PMO supplementation, as shown by flow cytometry analysis and confocal microscopy imaging (Fig. 2e, f and Supplementary Fig. 3a, b). RSL3 and erastin also increased total intracellular ROS, as shown by DCFH-DA staining followed by flow cytometry analysis. Similarly, PMO supplementation mitigated this increase in total intracellular ROS (Fig. 2g, h). Notably, PMO exposure failed to alter the transcription of endogenous antioxidant genes, including *Sod1* (superoxide dismutase 1), *Sod2* (superoxide dismutase 2), *Sod3* (superoxide dismutase 3), *Cat* (catalase), and *Txn* (thioredoxin) (Supplementary Fig. 3c), thus excluding the possibility that PMO inhibited ferroptosis by activating endogenous antioxidant systems.

Bioactive iron catalyzes the generation of lipid peroxides by supporting the Fenton reaction. Exposure to RSL3 and erastin increased labile iron, as shown by FerroOrange (the labile iron fluorescent probe) staining followed by flow cytometry analysis and confocal microscopy imaging, while PMO significantly decreased the fluorescence intensity of FerroOrange, suggesting that PMO reduced intracellular bioactive iron (Fig. 2i–l). GPX4 and FSP1 are two critical ferroptosis suppressors[5]. Cotreatment with PMO failed to prevent GPX4 degradation in response to RSL3 and erastin (Fig. 2m, n). Furthermore, PMO supplementation had a negligible effect on FSP1 expression under steady state and proferroptotic conditions (Fig. 2m, n). In addition, erastin triggers GSH exhaustion by inhibiting the cystine/glutamate antiporter System Xc⁻[5]. PMO administration failed to decrease GSH depletion in response to erastin exposure (Fig. 2o). Collectively, these data suggest that PMO can mimic multiple antioxidant enzymes and are potent ferroptosis inhibitors with lower cytotoxicity.

In addition to ferroptosis, apoptosis is another well-defined type of programmed cell death. Many kinds of chemotherapy drugs can induce apoptosis and slow tumor growth. PMO treatment failed to suppress apoptosis induced by Dox (doxorubicin), Eto (etoposide) and CPT (camptothecin), three important anticancer drugs (Supplementary Fig. 4).

## PMO are internalized by cells mainly *via* macropinocytosis

We then examined the underlying mechanism by which PMO counteracted ferroptosis. First, to evaluate PMO internalization, we labeled PMO with the fluorescent tracer DiI (1,1′-dioctadecy1−3,3,3′,3′-tetramethylindocarbocyanine perchlorate). The modified PMO were named PMO-DiI. Similar to nonfluorescent PMO, fluorescent PMO-DiI exhibited equivalent antiferroptotic capacity, as shown by microscopy imaging and cell viability assays (Supplementary Fig. 5a, b). Additionally, PMO-DiI relieved RSL3- and erastin-induced lipid peroxidation (Supplementary Fig. 5c, d), suggesting that coupling PMO with DiI did not change the antiferroptotic capacity of PMO.

We observed that PMO-DiI quickly entered cells (Fig. 3a, c). Fluorescence imaging and flow cytometry analysis showed that PMO uptake occurred in a time- and dose-dependent manner (Fig. 3a–d). It has been widely reported that nanoparticles are absorbed mainly

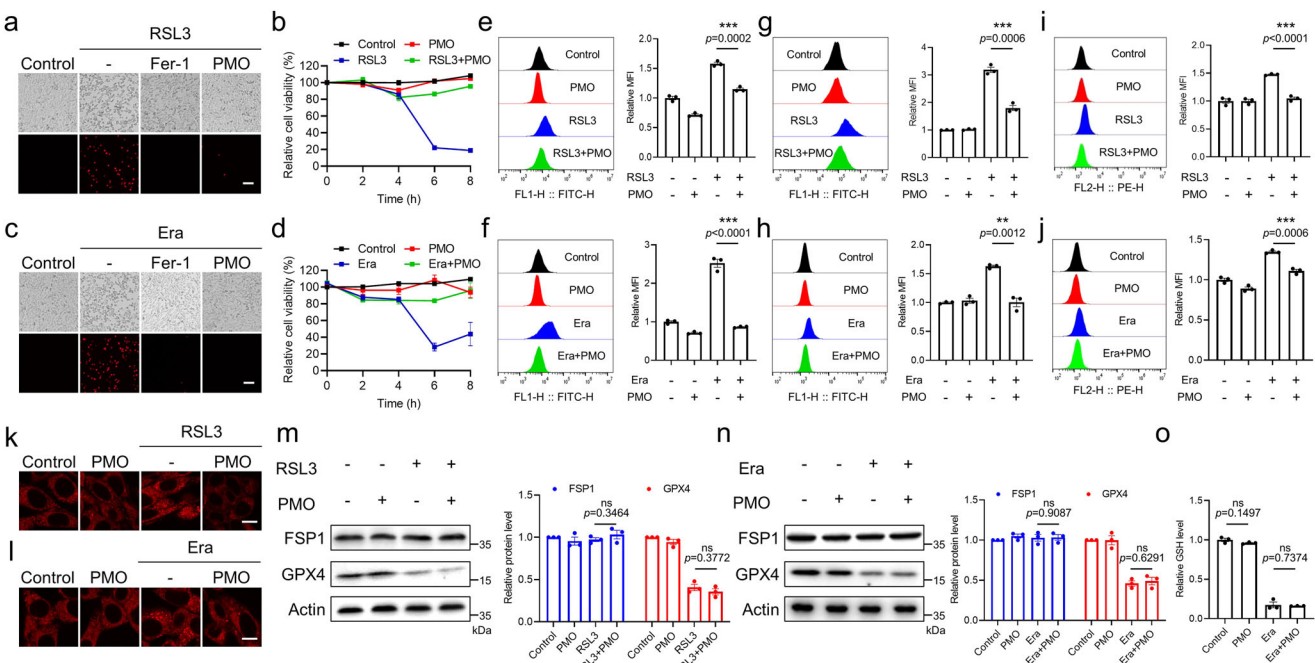

**Fig. 2 | PMO effectively counteract ferroptosis.** MEFs were treated with 1 μM RSL3, with or without 10 μM Fer-1 or 50 μg/mL PMO for the indicated time. Cell death was visualized by PI staining (**a**). Scale bar = 100 μm. The relative cell viability was analyzed by CCK-8 assay (**b**). MEFs were treated with 10 μM Era (Erastin), with or without 10 μM Fer-1 or 50 μg/mL PMO for the indicated time. Cell death was visualized by PI staining (**c**). Scale bar = 100 μm. The relative cell viability was analyzed by CCK-8 assay (**d**). **e**, **f** MEFs were treated as in (**b**) and (**d**). Lipid peroxidation was measured by BODIPY 581/591-C11 staining followed by flow cytometry analysis. **g**, **h** MEFs were treated as in (**b**) and (**d**). Total intracellular ROS was measured by DCFH-DA staining followed by flow cytometry analysis. **i**–**l** MEFs were treated as in (**b**) and (**d**). Intracellular bioactive iron was measured by FerroOrange staining followed by flow cytometry analysis (**i, j**) and confocal microscopy imaging (**k, l**). The staining was repeated three times independently with similar results. Scale bar = 10 μm. **m**, **n** MEFs were treated as in (**b**) and (**d**). The expressions of FSP1 and GPX4 were analyzed by Western blot. The relative protein levels were quantified. **o** MEFs were treated with 10 μM Era, with or without 50 μg/mL PMO for 4 h. GSH level was quantified. For statistical analysis, data represent mean ± SEM. $n = 4$ samples in **b, d**; $n = 3$ samples in **e**–**j, m**–**o**. ** $P < 0.01$, *** $P < 0.001$, ns $P > 0.05$, was determined by two-tailed unpaired Student's $t$ test. Source data are provided as a Source Data file.

through endocytosis, which can be categorized into clathrin-mediated endocytosis, caveolin-mediated endocytosis, macropinocytosis and others[27]. By utilizing the corresponding chemical inhibitors, we observed that exposure to Amiloride, a potent macropinocytosis inhibitor, greatly inhibited PMO-DiI uptake. Furthermore, inhibiting clathrin-mediated endocytosis with Dynasore resulted in an approximately 40% reduction in PMO-DiI uptake, while inhibiting caveolin-mediated endocytosis with Filipin had the least effect (Fig. 3e, f). Furthermore, Amiloride, but not Dynasore or Filipin, restored RSL3-induced ferroptosis in the presence of PMO, as shown by microscopy imaging and cell viability analysis (Fig. 3g, h). Collectively, these data indicate that PMO are internalized by cells mainly *via* macropinocytosis.

After internalization, the intracellular nanoparticles are corroded or stabilized. The amount of PMO biocorrosion was evaluated by measuring possible Mn release by ICP-MS (inductively coupled plasma mass spectrometry). PMO exposure failed to increase Mn concentrations in the cell culture medium or cell lysate (Fig. 3i), suggesting the excellent stability of PMO.

### Lysosome-enriched PMO inhibit ferritinophagy and protect lysosomes from lipid peroxidation

Confocal microscopy imaging suggested that intracellular PMO-DiI showed punctate distribution within cells (Fig. 3e). To identify the exact subcellular distribution of PMO-DiI, we labeled the major organelles, including lysosomes, mitochondria and ER (endoplasmic reticulum), and then incubated the cells with PMO-DiI. Confocal microscopy imaging showed that a large portion of PMO-DiI colocalized with LysoTracker, a fluorescent dye that labels lysosomes (Fig. 4a and Supplementary Fig. 6a). Additionally, the

colocalization of PMO-DiI with lysosomes was further confirmed by immunofluorescence staining of LAMP2 (lysosome-associated membrane protein 2), a typical marker protein of lysosomes. PMO-DiI largely colocalized with LAMP2 (Supplementary Fig. 6b), suggesting that PMO-DiI mainly resided in lysosomes. In contrast, PMO-DiI failed to colocalize with the ER or mitochondria (Supplementary Fig. 6c, d).

Lysosomes are one of the major organelles that determine cellular iron metabolism[28]. Specifically, lysosome-mediated ferritin degradation, which is known as ferritinophagy, complements the intracellular labile iron pool and expedites ferroptosis[14,15]. RSL3 and erastin exposure increased FTH (ferritin heavy chain) and decreased the ferritinophagy receptor NCOA4 (Fig. 4b, c). In addition, *Fth* mRNA was upregulated during ferroptosis induction (Fig. 4d, e). This finding was consistent with previous reports and could be explained by ferroptosis inducers initiating ferritinophagy for ferritin degradation and iron release, which could facilitate the transcriptional activation of *Fth* mRNA and the subsequent increase in FTH protein[14,15]. PMO supplementation inhibited the RSL3- and erastin-induced increase in FTH at both the protein and mRNA levels. In addition, PMO suppressed NCOA4 degradation, suggesting that PMO inhibited ferritinophagy (Fig. 4b–e). This finding was verified by intracellular bioactive iron analysis, which showed that PMO suppressed the RSL3- and erastin-induced increase in labile free iron (Fig. 2i–l). In addition, we evaluated ferritinophagy by measuring the translocation of ferritin to lysosomes. Exposure to RSL3 and erastin facilitated the colocalization of FTL (ferritin light chain) with the lysosome marker LAMP2, and this colocalization was dramatically suppressed by PMO supplementation (Fig. 4f, g and Supplementary Fig. 6e, f). The iron released by ferritinophagy is assumed to be primarily confined in lysosomes. We found

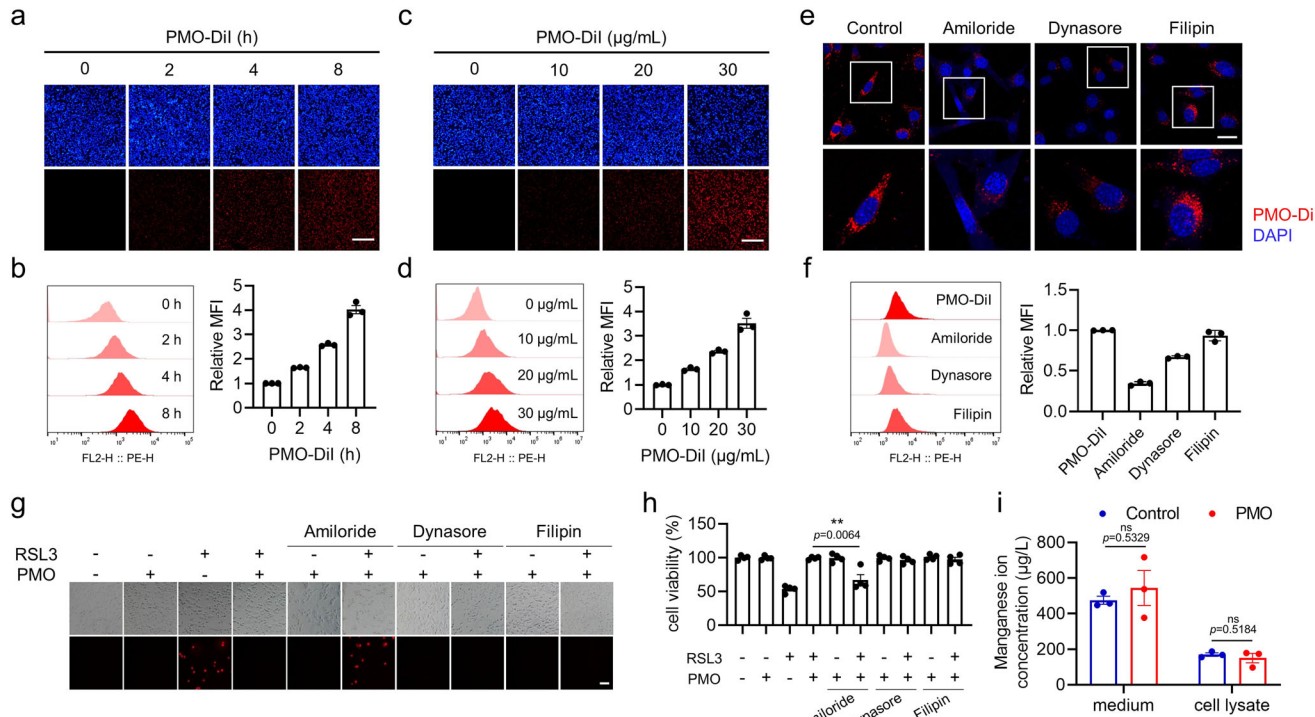

**Fig. 3 | PMO are internalized by cells mainly *via* macropinocytosis. a** MEFs were incubated with 20 µg/mL PMO-DiI for the indicated time. DAPI was used to label nucleus. Representative images were shown to suggest PMO-DiI uptake. Scale bar = 300 µm. **b** Cells were treated as in (**a**). Flow cytometry analysis was conducted to show PMO-DiI uptake. **c** MEFs were incubated with the indicated dose of PMO-DiI for 4 h. DAPI was used to label nucleus. Representative images were shown to suggest PMO-DiI uptake. Scale bar = 300 µm. **d** Cells were treated as in (**c**). Flow cytometry analysis was conducted to show PMO-DiI uptake. **e, f** MEFs were treated with the indicated inhibitors for 2 h, then incubated with 20 µg/mL PMO-DiI for 4 h. The cells were imaged by using a confocal microscope (**e**). Scale bar = 20 µm. The

cellular fluorescence was analyzed by flow cytometry (**f**). **g, h** MEFs were treated with the indicated inhibitors for 2 h, then incubated with 1 µM RSL3, with or without 50 µg/mL PMO for 4 h. Cell death was visualized by PI staining (**g**). Scale bar = 100 µm. The relative cell viability was measured by CCK-8 assay (**h**). **i** MEFs were incubated with 50 µg/mL PMO for 4 h. Mn concentrations in culture medium and cell lysates were analyzed by ICP-MS. For statistical analysis, data represent mean ± SEM. n = 3 samples in **b, d, f, i**; n = 4 samples in **h**. ** $P < 0.01$, ns $P > 0.05$, was determined by two-tailed unpaired Student's t test. Source data are provided as a Source Data file.

that RSL3 and erastin could increase lysosomal bioactive iron, as shown by the increase in FerroOrange colocalized with LysoTracker (Fig. 4h, i and Supplementary Fig. 6g, h). PMO supplementation greatly reduced lysosomal bioactive iron (Fig. 4h, i and Supplementary Fig. 6g, h). Taken together, these results indicate that PMO are enriched in lysosomes and inhibit ferritinophagy-mediated iron release.

How PMO suppress ferritinophagy was next examined. It has been reported that ROS are critical for ferritinophagy during ferroptosis induction[29]. We thus proposed that lysosome-resident PMO inhibited ferritinophagy by detoxifying ROS. First, lysosomal lipid peroxidation was measured by using Foma-LPO, a fluorescent lipid peroxidation probe that specifically targets lysosomes[30]. We observed an increase in Foma-LPO fluorescent signals, which colocalized with LysoTracker during RSL3 and erastin treatment. Lysosomal Foma-LPO fluorescence was sharply suppressed by PMO supplementation (Fig. 4j, k and Supplementary Fig. 6i, j), suggesting that PMO maintained lysosomal integrity and protected lysosomes from lipid peroxidation. In addition, cysteamine is an antioxidant that accumulates in lysosomes[31]. We found that cysteamine administration could potently inhibit ferroptosis (Supplementary Fig. 7a, b). More importantly, cysteamine could suppress ferritinophagy (Supplementary Fig. 7c, d). Therefore, it is assumed that PMO reside in lysosomes and function as antioxidants to scavenge lysosomal ROS, leading to the suppression of ferritinophagy in response to ferroptosis induction.

**PMO preserve mTOR activity to counteract ferroptosis**

After decades of being recognized as just a cellular recycling center, the lysosome has gradually been regarded as a critical signaling hub[32].

Specifically, mTOR, a major signaling axis responsible for ferroptosis regulation, is dynamically activated on the lysosomal membrane[32]. To further elucidate the underlying mechanism by which lysosome-resident PMO inhibit ferroptosis in addition to suppressing ferritinophagy-mediated iron mobilization, mTOR activity was examined by analyzing the phosphorylation level. RSL3 and erastin inactivated mTOR, as evidenced by the decrease in mTOR phosphorylation, while PMO supplementation restored mTOR activation. However, PMO failed to preserve mTOR activity in the presence of the mTOR inhibitor Torin (Fig. 5a, b). To investigate whether the preservation of mTOR activity is responsible for PMO-mediated inhibition of ferroptosis, the mTOR inhibitor Torin was used. Torin alone resulted in minimal cytotoxicity. When administered with Torin, RSL3- and erastin-induced ferroptosis occurred in the presence of PMO (Fig. 5c, d). More importantly, Torin restored the accumulation of lipid peroxides induced by RSL3 and erastin, which was mitigated by PMO supplementation (Fig. 5e, f).

Collectively, these results demonstrate that PMO inhibit ferroptosis through multifaceted mechanisms. By being enriched in lysosomes, PMO detoxify lysosomal ROS, suppress lysosomal ferritinophagy to reduce lysosomal and intracellular bioactive iron, and preserve mTOR activation, ultimately suppressing ferroptosis.

**Intravenously injected PMO are enriched in the liver and show good biocompatibility**

The excellent antiferroptotic capacity in vitro prompted us to explore the potential application of PMO to alleviate ferroptosis-associated

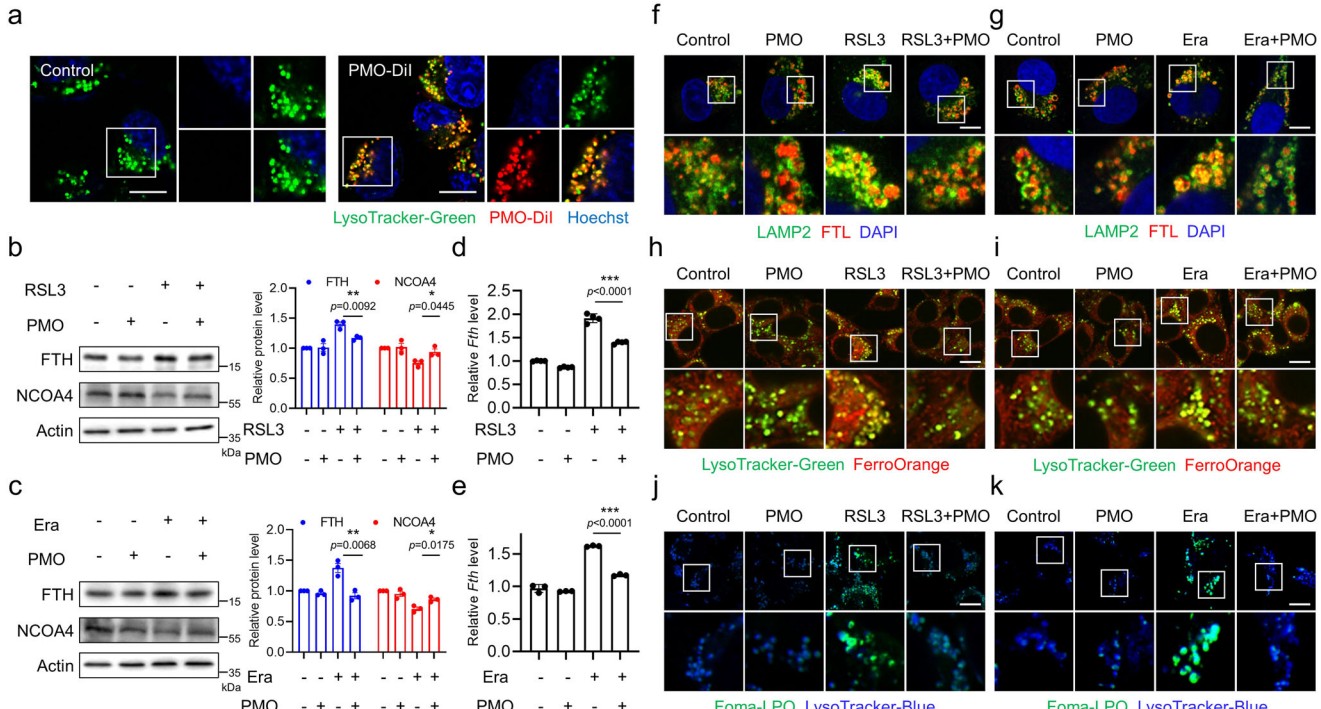

**Fig. 4 | Lysosome-enriched PMO inhibit ferritinophagy and protect lysosomes from lipid peroxidation. a** MEFs were incubated with 20 µg/mL PMO-DiI for 4 h. Lysosomes and nucleus were labeled by LysoTracker-Green and Hoechst, respectively. Images were captured by using a confocal microscope. Scale bar = 10 µm. The staining was repeated three times independently with similar results. **b, c** MEFs were treated with 1 µM RSL3 (**b**) or 10 µM Era (**c**), with or without 50 µg/mL PMO. The expressions of FTH and NCOA4 were analyzed by Western blot. The relative protein levels were quantified. **d, e** MEFs were treated as in (**b**) and (**c**). Real-time PCR was carried out to check *Fth* transcription. **f, g** Cells were treated as in (**b**) and (**c**). The cells were fixed and stained with anti-FTL and anti-LAMP2 antibodies.

Images were captured by using a confocal microscope. Scale bar = 10 µm. **h, i** Cells were treated as in (**b**) and (**c**). The cells were stained with LysoTracker-Green and FerroOrange. Images were captured by using a confocal microscope. Scale bar = 10 µm. **j, k** Cells were treated as in (**b**) and (**c**). The cells were stained with LysoTracker-Blue and Foma-LPO. Images were captured by using a confocal microscope. Scale bar = 10 µm. The staining in **f–k** was repeated three times independently with similar results. For statistical analysis, data represent mean ± SEM. n = 3 samples in **b, c, e**; and n = 4 samples in **d**. * $P < 0.05$, ** $P < 0.01$, *** $P < 0.001$, was determined by two-tailed unpaired Student's t test. Source data are provided as a Source Data file.

pathogenesis in vivo. First, we intravenously injected PMO into mice, and the major tissues were isolated 1, 14 and 30 days postinjection. The normal histology of the tissues, including heart, liver, spleen, lung and kidney, suggested good biocompatibility of PMO (Fig. 6a). This was further confirmed by serological tests. PMO injection induced negligible changes in serum AST and ALT (aspartate transaminase and alanine transaminase, respectively, two classic indicators of liver damage), BUN and CR (blood urea nitrogen and creatinine, respectively, two renal function indicators), and CK-MB and LDH (creatine kinase-MB and lactate dehydrogenase, respectively, two cardiac injury markers) (Fig. 6b).

Next, the biodistribution of PMO was evaluated by measuring Mn levels in major tissues 1, 14 and 30 days postinjection of PMO by using ICP-MS. The results showed that PMO injection increased Mn levels in the liver 1 day postinjection, suggesting that PMO were primarily distributed in the liver (Fig. 6c). On days 14 and 30 postinjection, Mn concentrations in the liver were reduced to normal levels (Fig. 6c). Moreover, intravenous PMO injection failed to increase Mn levels in other tissues, including heart, spleen, lung and kidney, suggesting that PMO did not accumulate in these tissues (Fig. 6c). In addition, the biodistribution of PMO was confirmed by the injection of fluorescent PMO-DiI. Bioluminescence imaging indicated that PMO-DiI were enriched in the liver 1 day postinjection, but not in other tissues (Supplementary Fig. 8). Moreover, fluorescence imaging showed that PMO-DiI fluorescence was mainly enriched in the liver 1 day postinjection and then vanished over time (Fig. 6d). Similarly, fluorescence imaging showed that PMO-DiI did not target heart, spleen, lung or kidney (Fig. 6d).

## PMO alleviate drug-induced liver injury

Ferroptosis is implicated in a variety of liver diseases. Notably, APAP overdose leads to serious liver injury, during which the typical ferroptosis characteristics are observed[17,18]. To investigate the protective effect of PMO against drug-induced liver injury, a mouse model of APAP-induced liver injury was established by intraperitoneal injection of APAP. The mice were intravenously injected with PMO (20 mg/kg bodyweight) twice, 24 h before APAP administration and simultaneously with APAP injection. Hepatic injury was evaluated 24 h after APAP injection (Fig. 7a). Serum biochemical analysis showed that PMO injection decreased serum ALT and AST, two major indicators of liver damage (Fig. 7b, c), suggesting a protective effect of PMO against APAP-induced liver injury. This finding was confirmed by H&E and TUNEL (terminal deoxynucleotidyl transferase-mediated deoxyuridine triphosphate nick end labeling) staining. PMO injection significantly decreased the necrotic areas in the livers of mice with APAP intoxication (Fig. 7d–g).

Given the pathological role of ferroptosis in APAP-induced liver injury, ferroptosis characteristics were further investigated. 4-HNE (4-hydroxynonenal) and MDA (malondialdehyde) are two aldehyde secondary products of lipid peroxides and are widely used as biomarkers of lipid peroxidation. We found that 4-HNE and MDA levels were significantly increased in response to APAP injection and could be suppressed by PMO (Fig. 7h, i). Lipid peroxides and secondary products can attack DNA, leading to the generation of 8-OHdG (8-hydroxydeoxyguanosine) adducts, which are markers of DNA oxidative damage. APAP exposure increased the 8-OHdG signal, and this effect was largely mitigated by PMO, as evidenced by immunofluorescence

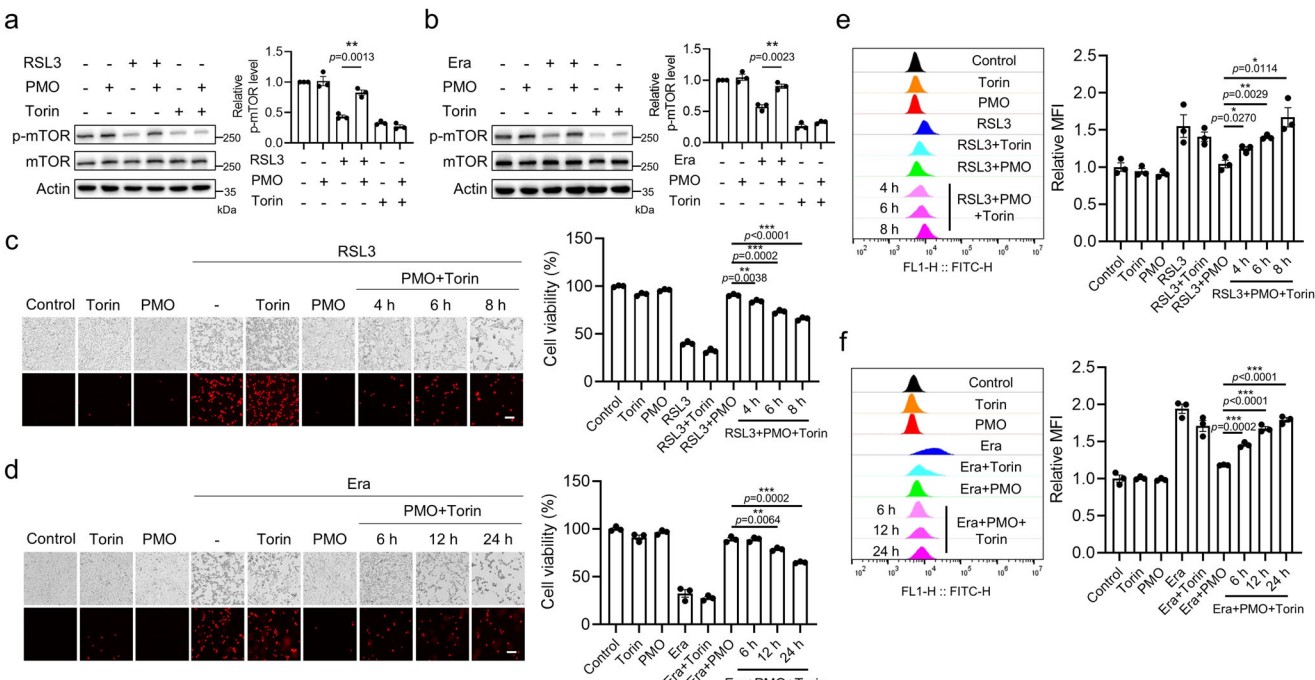

**Fig. 5 | PMO preserve mTOR activity to counteract ferroptosis. a, b** MEFs were treated with 5 μM Torin and 1 μM RSL3 (**a**) or 10 μM Era (**b**), with or without 50 μg/mL PMO. The total and phosphorylated mTOR were analyzed by Western blot. The phosphorylated mTOR normalized to the total mTOR was shown in the histograms. **c, d** MEFs were treated with 5 μM Torin and 1 μM RSL3 (**c**) or 10 μM Era (**d**), with or without 50 μg/mL PMO for the indicated time. Cell death was visualized by PI staining. Scale bar = 100 μm. The relative cell viability was measured by CCK-8 assay and shown in the histograms. **e, f** Cells were treated as in (**c**) and (**d**). Lipid peroxidation was measured by BODIPY 581/591-C11 staining followed by flow cytometry analysis. For statistical analysis, data represent mean ± SEM. $n$ = 3 samples in **a**–**f**. * $P < 0.05$, ** $P < 0.01$, *** $P < 0.001$, ns $P > 0.05$, was determined by two-tailed unpaired Student's t test. Source data are provided as a Source Data file.

staining using a specific anti-8-OHdG antibody (Fig. 7j). *Ptgs2* (prostaglandin-endoperoxide synthase 2) is typically considered a biomarker of ferroptosis[6]. APAP intoxication led to a significant increase in *Ptgs2* mRNA, while PMO administration suppressed this increase (Fig. 7k). More importantly, we found that PMO reduced liver iron (Fig. 7l), and suppressed the APAP-induced increase in FTH (Fig. 7m). We also found that PMO counteracted APAP-induced mTOR dephosphorylation (Fig. 7n). Overall, these mouse studies suggest that PMO injection alleviates APAP-induced liver injury, at least partially by counteracting ferroptosis.

We also verified the antiferroptotic capacity of PMO in hepatocytes. Cell viability assays showed that PMO administration potently inhibited RSL3- and erastin-induced ferroptosis in AML12 mouse hepatocytes (Supplementary Fig. 9a, b). Furthermore, PMO administration mitigated APAP-induced ferroptosis in AML12 mouse hepatocytes (Supplementary Fig. 10a), and RSL3-, erastin- and APAP-induced ferroptosis in L02 human hepatocytes (Supplementary Fig. 10b-d). PMO also suppressed lipid peroxidation, which was increased by RSL3 and erastin in mouse hepatocytes (Supplementary Fig. 9c, d). Likewise, PMO administration failed to alter the expression of GPX4 or FSP1 (Supplementary Fig. 9e, f). Bioactive iron analysis suggested that PMO decreased the labile iron pool in mouse hepatocytes (Supplementary Fig. 9g, h). PMO-DiI entered mouse hepatocytes in a time- and dose-dependent manner (Supplementary Fig. 9i, j), and the intracellular PMO-DiI mainly colocalized with lysosomes, as shown by confocal microscopy imaging (Supplementary Fig. 9k). In addition, we observed that PMO could suppress ferritinophagy in mouse hepatocytes, as evidenced by the reduction in the protein and mRNA levels of FTH, and the decrease in NCOA4 degradation during ferroptosis induction (Supplementary Fig. 9l-o). In this context, PMO administration potently reduced lipid peroxidation in lysosomes, as shown by costaining with LysoTracker and Foma-LPO, a specific dye used to label lysosomal lipid peroxidation (Supplementary Fig. 9p, q). Moreover,

PMO preserved mTOR phosphorylation in ferroptotic mouse hepatocytes (Supplementary Fig. 9r, s). These results suggest that PMO are potent inhibitors of ferroptosis in vitro and in vivo.

## PMO alleviate liver ischemia/reperfusion injury

LIRI (liver ischemia/reperfusion injury) is a major cause of liver failure following liver resection and liver transplantation[33]. An early study suggested that the ferroptosis inhibitor could reduce hepatic damage in mice with LIRI, indicating that ferroptosis is implicated in the pathogenesis of LIRI[34]. To further determine the potential application of PMO to protect mice from LIRI, we intravenously injected PMO and then established a mouse LIRI model (Fig. 8a). Serum biochemical analysis showed that PMO injection reduced serum ALT and AST (Fig. 8b, c). PMO injection decreased the necrotic areas, as shown by H&E staining (Fig. 8d, e). TUNEL staining further confirmed the protective effect of PMO against LIRI (Fig. 8f, g). In addition, we evaluated the ferroptosis levels in these mice by measuring the secondary products of lipid peroxidation, including 4-HNE and MDA. Hepatic 4-HNE and MDA levels were increased during LIRI, while PMO injection significantly suppressed these increases (Fig. 8h, i). Immunofluorescence staining showed that PMO injection reduced hepatic 8-OHdG adducts (Fig. 8j). More importantly, PMO administration notably decreased the transcription of *Ptgs2* (Fig. 8k), a widely used ferroptosis biomarker. Tissue iron analysis showed that PMO significantly decreased hepatic iron (Fig. 8l). Furthermore, Western blot analysis showed that PMO injection decreased FTH levels and preserved mTOR phosphorylation under LIRI (Fig. 8m, n). These results suggest that intravenous injection of PMO protects mice from LIRI.

Excessive inflammatory response occurs during LIRI. Notably, macrophage and neutrophil infiltration induces local inflammation and expedites hepatic injury[33]. LIRI increased the expression of several chemokines, including *Ccl5*, *Cxcl1* and *Cxcl2* (Supplementary Fig. 11a-c), which encode important regulators that recruit macrophages and

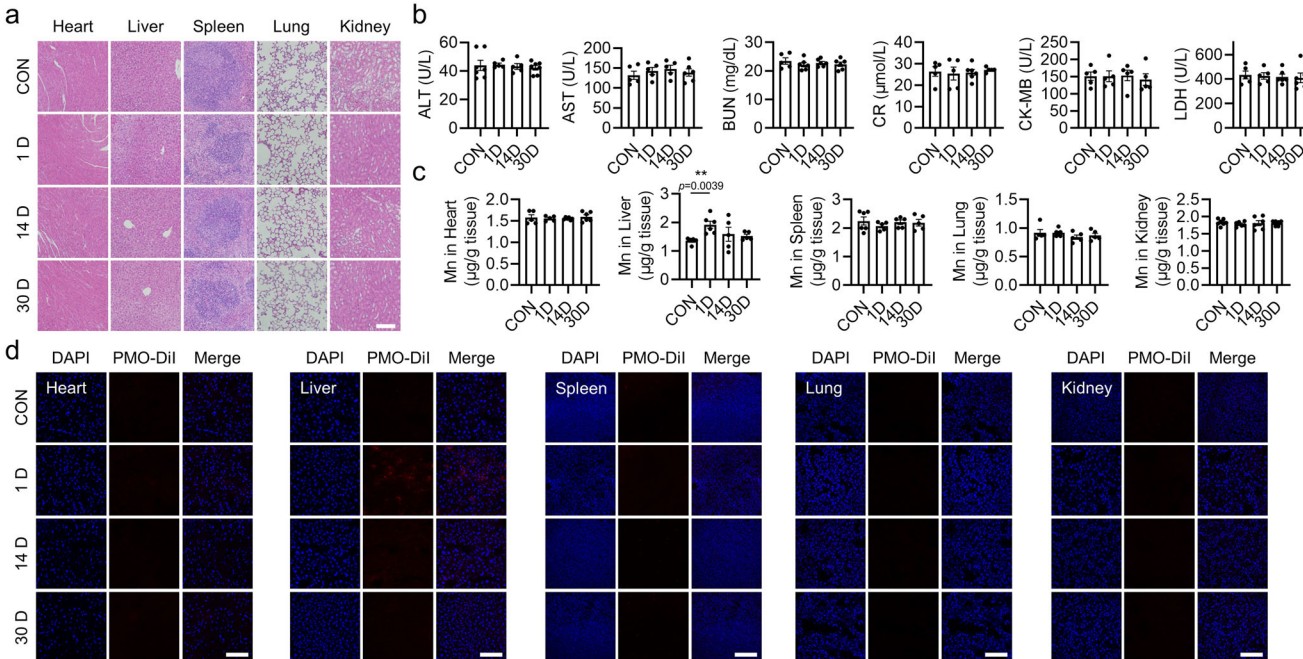

**Fig. 6 | Intravenously injected PMO are enriched in the liver and show good biocompatibility. a** C57BL/6 J mice were intravenously injected with PMO at 20 mg/kg bodyweight. The mice were sacrificed 1, 14 and 30 days postinjection, and the indicated tissues were isolated for H&E staining. Scale bar = 100 μm. **b** Serum biochemical analysis of AST, ALT, BUN, CR, CK-MB, and LDH. **c** Mn concentrations in the indicated tissues. **d** C57BL/6 J mice were intravenously injected with PMO-DiI at 20 mg/kg bodyweight. The mice were sacrificed 1, 14 and 30 days postinjection, and the indicated tissues were isolated for fluorescence microscopy imaging. Scale bar = 100 μm. For statistical analysis, data represent mean ± SEM. In **b** and **c**, n = 7 independent mice in control group and 30 days group, n = 5 independent mice in the 1 day group and 14 days group. ** P < 0.01, was determined by two-tailed unpaired Student's t test. Source data are provided as a Source Data file.

neutrophils. In addition, the proinflammatory cytokine *Il-1β* was upregulated in LIRI models (Supplementary Fig. 11d). These chemokines and the cytokine *Il-1β* were decreased at the mRNA level by PMO injection (Supplementary Fig. 11), suggesting that PMO alleviate LIRI-induced hepatic inflammation.

## Discussion

It has been gradually recognized that ferroptosis is involved in the pathogenesis of various liver diseases, and targeted inhibition of ferroptosis holds great promise for mitigating the progression of liver damages[35]. In this study, we report that Mn-based PMO nanoparticles mimic the activities of multiple antioxidant enzymes and can scavenge broad-spectrum ROS, especially OH•. After being absorbed mainly through macropinocytosis, PMO reside in lysosomes, where PMO detoxify ROS, suppress ferritinophagy and preserve mTOR phosphorylation. Through these mechanisms, PMO potently inhibit ferroptosis. Furthermore, intravenously injected PMO show good biocompatibility and could significantly alleviate APAP-induced and hepatic ischemia/reperfusion-induced liver injury (Fig. 9).

The ROS scavenging assays showed that PMO exhibit CAT-like $H_2O_2$ scavenging activity and SOD-like $O_2^{•-}$ scavenging activity. In addition, PMO perform potent scavenging activity for OH•, which cannot be directly detoxified by certain endogenous antioxidant enzymes. Our other experiment showed that PMO-mediated ROS scavenging is exothermic and can occur automatically. This is related to the exposed 101 facets and the $Mn^{2+}/Mn^{3+}/Mn^{4+}$ ratio (0.4:1:1.9) of PMO[36]. A previous study by Singh and colleagues suggested that $Mn_3O_4$ nanozymes exhibit SOD-, CAT- and GPX-like enzymatic activities in a size- and morphology-dependent manner[37]. This finding was confirmed by a subsequent study[38] and the current study. Furthermore, the ROS scavenging assays indicated that PMO (50–200 μg/mL) exhibit higher scavenging activity for OH• than for $H_2O_2$ or $O_2^{•-}$. Since OH• is the indeed ROS that directly attacks PUFAs to generate lipid peroxides, the detoxification of OH• ensures the powerful inhibition of ferroptosis by PMO.

PMO absorption is dependent on macropinocytosis and clathrin-mediated endocytosis, but not caveolin-mediated endocytosis. Supplementation of macropinocytosis inhibitor notably restored RSL3-induced ferroptosis in the presence of PMO, suggesting that macropinocytosis is the main pathway mediating PMO uptake. Macropinocytosis is a regulated form of endocytosis that mediates the nonselective uptake of extracellular molecules, nutrients, antigens and other particles. The surrounding substance is engulfed and internalized by macropinosomes, which are formed through actin-dependent membrane ruffling and protrusion across the cell membrane[39]. Previous studies demonstrated that compared to clathrin- and caveolin-mediated endocytosis, macropinocytosis could mediate the uptake of substances with a wider range of sizes, due to the large sizes of macropinosomes (>200 nm)[40].

After internalization, PMO are enriched in lysosomes. Lysosomes contain acid hydrolases and serve as digestive organelles for the turnover of intracellular components or endocytic substances. Recently, independent studies suggested that lysosomal ferritinophagy is indispensable for the mobilization of ferritin-bound iron and thus important for the execution of ferroptosis. Genetic depletion of key autophagy genes or the specific ferritinophagy receptor NCOA4 reduces intracellular bioactive iron and desensitizes cells to ferroptosis[14,15]. PMO suppressed ferritinophagy, as evidenced by the decrease in ferritin turnover, reduction in lysosomal translocation of ferritin, and decrease in intracellular and lysosomal iron. A previous study suggested that ROS are indispensable for ferritinophagy[29]. Therefore, it is assumed that PMO inhibit ferritinophagy by detoxifying excessive ROS. In addition, lysosomes have gradually been regarded as signaling hubs that orchestrate signal transduction in response to intracellular and extracellular stimuli[41]. mTOR, the master metabolic regulator that is activated on the lysosomal surface, dictates ferroptosis vulnerability by determining the cellular lipidomic signature[42–46]. Lysosome-resident PMO preserved mTOR activation under ferroptotic conditions. Chemical inhibition of mTOR restored erastin- and RSL3-

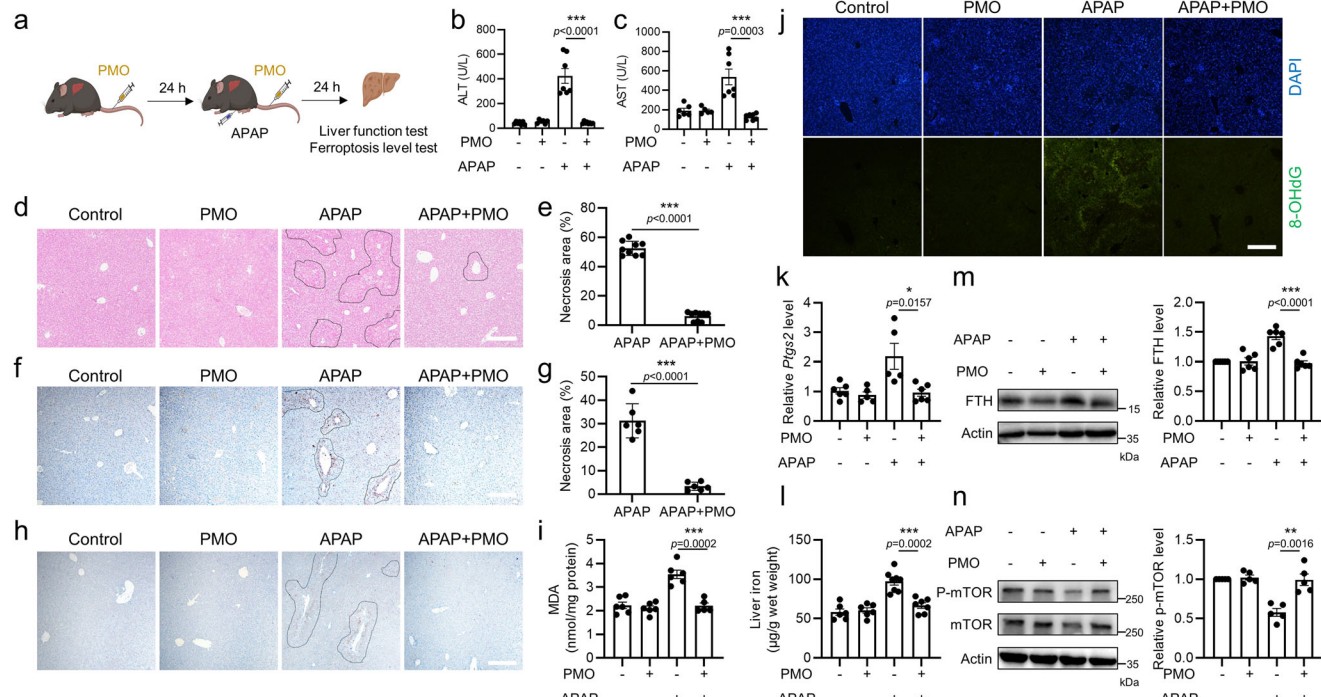

**Fig. 7 | PMO alleviate drug-induced liver injury. a** Treatment schedule of APAP-induced liver injury and PMO intervention in mice. **b, c** Serum ALT (**b**) and serum AST (**c**) in the mice. **d** H&E staining of the liver tissues. Scale bar = 100 μm. Black dashed lines indicate the necrotic areas. **e** The percentages of necrotic areas in (**d**) were quantified. **f** TUNEL staining of the liver tissues. Scale bar = 100 μm. Black dashed lines indicate the TUNEL-positive areas. **g** The percentages of TUNEL-positive areas in (**f**) were quantified. **h** Immunohistochemical staining of 4-HNE in the liver tissues. Scale bar = 100 μm. **i** MDA concentrations in the liver tissues. **j** Immunofluorescence staining of 8-OHdG in the liver tissues. Scale bar = 100 μm. **k** mRNA level of *Ptgs2* in the liver tissues. **l** Liver iron in the mice. **m, n** FTH, as well as the total and phosphorylated mTOR were analyzed by Western blot. The relative protein levels were quantified. For statistical analysis, data represent mean ± SEM. In **b**, $n = 7$ mice in control group and APAP group, $n = 6$ mice in PMO group, n = 8 mice in APAP + PMO group. In c, n = 6 mice in control group, $n = 5$ mice in PMO group, n = 7 mice in APAP group and APAP + PMO group. In e, $n = 9$ mice in each group. In g, n = 6 mice in each group. In **i**, $n = 6$ mice in each group. In **k**, n = 6 mice in control group and APAP + PMO group, $n = 5$ mice in PMO group and APAP group. In **l**, n = 6 mice in control group and PMO group, $n = 8$ mice in APAP group, n = 7 mice in APAP + PMO group. In **m**, n = 6 mice in each group. In **n**, $n = 5$ mice in each group. * $P < 0.05$, ** $P < 0.01$, *** $P < 0.001$, was determined by two-tailed unpaired Student's t test. Source data are provided as a Source Data file.

induced ferroptosis in the presence of PMO, suggesting that PMO inhibit ferroptosis by maintaining mTOR activation. Excessive ROS trigger mTOR inactivation[47,48]. Thus, we propose that PMO preserve mTOR activation by detoxifying ROS in lysosomes during ferroptosis induction.

Histological analysis and serological tests suggested that the intravenously injected PMO showed good biocompatibility. Notably, PMO were specifically enriched in the liver after intravenous injection, although the underlying mechanism is unclear. Ferroptosis is implicated in the pathogenesis of several kinds of liver diseases, including drug-induced liver injury and liver ischemia/reperfusion injury[5]. APAP overdose is the leading cause of drug-induced liver injury[49]. It is estimated that 30,000 patients are admitted to intensive care units every year in America due to APAP overdose-induced liver injury[50]. To date, the ROS scavenger NAC (N-acetyl cysteine) is the only effective antidote against APAP-induced liver injury approved by the Food and Drug Administration[51]. Additionally, liver ischemia/reperfusion injury is a major cause of liver damage during surgical procedures, including liver resection and liver transplantation[52]. Mitigating liver ischemia/reperfusion injury would improve the clinical outcomes of these liver surgeries[33]. However, there is no approved pharmacological treatment available currently[53]. Lipid peroxidation, the typical hallmark of ferroptosis, was observed in the mouse models of APAP-induced acute liver injury and liver ischemia/reperfusion injury. PMO injection significantly alleviated liver damage, as evidenced by reduced serum ALT and AST, as well as decrease in hepatocyte death. More importantly, PMO injection counteracted lipid peroxidation and mitigated

hepatocyte ferroptosis. Therefore, PMO provide striking protection in the liver against APAP intoxication and ischemia/reperfusion injury by mitigating ferroptosis.

In summary, our results suggest that the biocompatible PMO nanoparticles can mimic the activities of multiple antioxidant enzymes and robustly inhibit ferroptosis through multifaceted mechanisms. PMO are promising candidates for clinical treatment of APAP-induced acute liver injury. Furthermore, by alleviating ferroptosis-associated liver ischemia/reperfusion injury, PMO hold great promise for improving the clinical outcomes of liver surgeries, including liver resection and liver transplantation.

## Methods
### Mice and research compliances
8-week-old male C57BL/6 J mice were purchased from the laboratory animal center of Huazhong Agricultural University, Wuhan, China. Mice were housed in a specific pathogen free animal facility at 24 °C with a 12-h light/12-h dark cycle and 40–60% humidity. Mice were given free access to water and food. The animal experiments were performed according to the procedures approved by the Laboratory Animal Welfare and Ethics Committee of Huazhong Agricultural University (No. HZAUMO-2022-0133).

### Reagents
RSL3 (S8155), erastin (S7242), Fer-1 (S7243) and cysteamine (S4206) were purchased from Selleck. Torin (HY-13003), Etoposide (HY-13629), Doxorubicin (HY-15142A), Camptothecin (HY-16560), Filipin (HY-

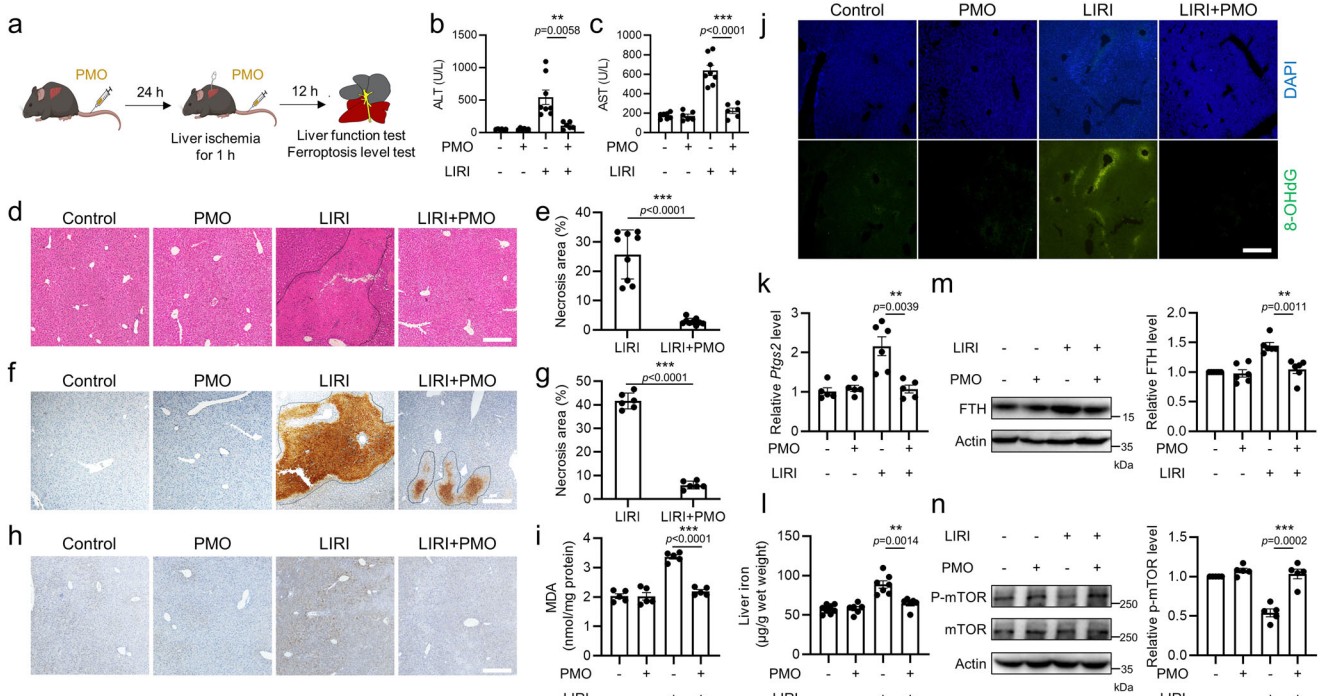

**Fig. 8 | PMO alleviate liver ischemia/reperfusion injury. a** Treatment schedule of hepatic ischemia/reperfusion-induced liver injury and PMO intervention in mice. **b, c** Serum ALT (**b**) and serum AST (**c**) in the mice. **d** H&E staining of the liver tissues. Scale bar = 100 μm. Black dashed lines indicate the necrotic areas. **e** The percentages of necrotic areas in (**d**) were quantified. **f** TUNEL staining of the liver tissues. Scale bar = 100 μm. Black dashed lines indicate the TUNEL-positive areas. **g** The percentages of TUNEL-positive areas in (**f**) were quantified. **h** Immunohistochemical staining of 4-HNE in the liver tissues. Scale bar = 100 μm. **i** MDA concentrations in the liver tissues. **j** Immunofluorescence staining of 8-OHdG in the liver tissues. Scale bar = 100 μm. **k** mRNA level of *Ptgs2* in the liver tissues. **l** Liver iron in the mice. **m, n** FTH, as well as the total and phosphorylated mTOR were analyzed by Western

blot. The relative protein levels were quantified. For statistical analysis, data represent mean ± SEM. In **b**, n = 6 mice in control group, PMO group and LIRI + PMO group, n = 8 mice in LIRI group. In **c**, n = 7 mice in control group, n = 6 mice in PMO group and LIRI + PMO group, n = 8 mice in LIRI group. In **e**, n = 9 mice in each group. In **g**, n = 6 mice in each group. In **i**, n = 5 mice in each group. In **k**, n = 5 mice in control group, PMO group and LIRI + PMO group, n = 6 mice in LIRI group. In **l**, n = 8 mice in control group, n = 6 mice in PMO group and LIRI + PMO group, n = 7 mice in LIRI group. In **m**, n = 6 mice in each group. In **n**, n = 5 mice in each group. ** $P < 0.01$, *** $P < 0.001$, was determined by two-tailed unpaired Student's t test. Source data are provided as a Source Data file.

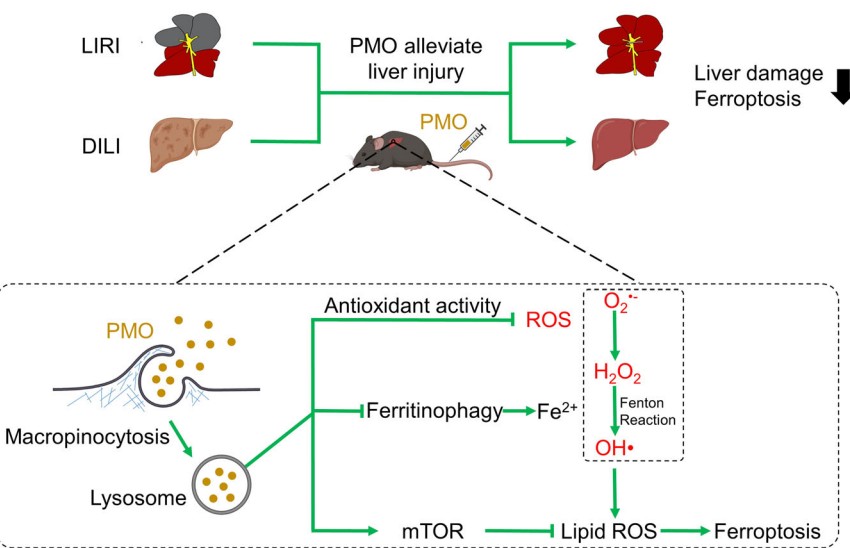

**Fig. 9 | Schematic illustration of PMO in counteracting ferroptosis and alleviating liver injury.** PMO can mimic the activities of multiple antioxidant enzymes and scavenge broad-spectrum ROS. After being absorbed mainly through macropinocytosis, PMO largely reside in lysosomes, where PMO scavenge ROS, inhibit

ferritinophagy-mediated iron release and preserve mTOR activation, leading to the robust inhibition of ferroptosis. The intravenously injected PMO are mainly enriched in the liver and can alleviate DILI (drug-induced liver injury) and LIRI (liver ischemia/reperfusion injury).

N6716), Amiloride (HY-B0285A) and Dynasore (HY-15304) were purchased from MedChem Express.

The primary antibodies used in this study are as follows: anti-GPX4 rabbit monoclonal antibody (ab125066) and anti-4-HNE goat polyclonal antibody (ab46544) were purchased from Abcam. anti-FSP1 rabbit polyclonal antibody (20886-1-AP), anti-FTL rabbit polyclonal antibody (10727-1-AP) and anti-Actin mouse monoclonal antibody (60008-1-Ig) were purchased from Proteintech. anti-mTOR (2983 S), anti-p-mTOR (5536 S) and anti-FTH (4393 S) rabbit monoclonal antibodies were purchased from Cell Signaling Technology. anti-LAMP2 (SC-18822), anti-NCOA4 (SC-373739) and anti-8-OHdG (SC-66036) mouse monoclonal antibodies were purchased from Santa Cruz Biotechnology. The HRP-conjugated secondary antibodies used in this study include anti-mouse IgG (7076 S) and anti-rabbit IgG (7074 S), which were purchased from Cell Signaling Technology. The fluorescent secondary antibodies used in this study include goat anti-mouse IgG Alexa Fluor™ 488 (A11029) and goat anti-rabbit IgG Alexa Fluor™ 594 (A11037), which were purchased from Thermo Fisher Scientific. The detailed information of antibodies used in this study are provided in Supplementary Table 1.

## Synthesis and characterization of PMO
The synthesis of PMO was conducted according to the following method. Briefly, dissolve 0.425 g MnSO$_4$·H$_2$O (M7634, Sigma Aldrich) into 2.5 mL deionized water and 4.5 g poly(acrylic) acid (323667, Sigma Aldrich) into 5.0 mL deionized water. Use a vortex mixer (VORTEX-7) to mix the two solutions at 1000 x g for 15 min. Then, the mixture was dropwisely added into a glass beaker (50 mL) containing 15 mL ammonium hydroxide (30%, 05002, Sigma Aldrich). After overnight stirring the mixture at 300 x g under room temperature, the solution was added into a Teflon-equipped stainless autoclave (50 mL). Heat the autoclave at 120 °C for 24 h. Centrifuge the resulted brown solution at 1520 x g for 1 h to remove the possible debris and large agglomerates. After taking the supernatant, purify it through dialysis for 24 h (refresh the deionized water once every 8 h) to collect PMO. Then, filter the purified PMO suspension through a syringe filter with 100-nm-pore size (BIOFIL). Store the obtained PMO solution in 4 °C refrigerator for further use. PMO were characterized following the methods described in our previous paper[54]. Briefly, the hydrodynamic diameter and zeta potential of PMO were measured by a Zeta-sizer (NanoBrook90 Plus, Brookhaven). TEM images were captured by using an FEI Talos microscope operating at 200 kV. FTIR spectroscopy was done by using an Avatar-330 spectrometer (Thermo Fisher, USA).

## H$_2$O$_2$ scavenging activity assay
H$_2$O$_2$ scavenging activity of PMO was detected by monitoring the characteristic absorbance of H$_2$O$_2$ at 240 nm under PMO administration. PMO were diluted into different concentrations and mixed with H$_2$O$_2$. The mixtures were placed for 30 min in the dark, then the absorbance of the mixtures at 240 nm was monitored by using a UV-vis spectroscopy. H$_2$O$_2$ scavenging activities of PMO were calculated as follows:

$$H_2O_2 \, \text{scavenging ratio} \, (\%) = ((A_0 - (A_1 - A_2))/A_0) \times 100$$

where $A_O$ is the absorbance of H$_2$O$_2$ solution without PMO. $A_1$ is the absorbance of H$_2$O$_2$ solution after the incubation with PMO for 30 min. $A_2$ is the absorbance of PMO diluent.

## OH• scavenging activity assay
OH• scavenging activity of PMO was assessed by using the methyl violet (MV)-Fenton reagent system, according to a previous report[55]. OH• could fade purple MV to a colorless appearance with decreased absorbance of MV at 582 nm. The different concentrations of PMO were prepared by gradient dilution of PMO stock with 0.1 M Tris-HCl buffer at pH 4.7. OH• was generated through classical Fenton reaction with 1.8 mM FeSO$_4$ and 0.1 M H$_2$O$_2$. After incubation for 10 min at room temperature, the absorbance of the solution at 582 nm was monitored by using a UV-vis spectroscopy. OH• scavenging activities of PMO were calculated as follows:

$$OH• \, \text{scavenging ratio} \, (\%) = ((A_1 - A_0)/(A_2 - A_0)) \times 100$$

where $A_O$ is the absorbance of MV in Fenton system without PMO. $A_1$ is the absorbance of MV in Fenton system after incubation with different concentrations of PMO. $A_2$ is the absorbance of MV.

## O$_2$•⁻ scavenging activity assay
O$_2$•⁻ scavenging activity of PMO was conducted according to the nitroblue tetrazolium colorimetric method by using a commercial kit, which was purchased from Beijing Solarbio Science & Technology (BC1415).

## Cell culture
MEFs, HT1080, AML12 and L02 cells were obtained in the lab. Cells were maintained in DMEM (Dulbecco's modified Eagle's medium) supplemented with 10% FBS (fetal bovine serum) and 1% (v/v) penicillin/streptomycin in an incubator supplied with a humidified atmosphere of 5% CO$_2$ at 37 °C.

## Cell viability and cell death assay
Cell viability was measured by CCK-8 assay using a commercial kit (Cell Counting Kit-8, C0038, Beyotime Biotechnology). Briefly, 10000 cells per well were seeded in 96-well plates and treated as indicated in each experiment. Subsequently, the culture medium was discarded, and 100 μL CCK-8 working solution (10 μL CCK-8 reagent diluted in 90 μL culture medium) was added to each well. The plate was incubated at 37 °C for 1 h. The absorbance of each well was measured with a microplate reader at 450 nm. The relative cell viability was normalized to the control group. Cell death analysis was performed by PI (HY-D0815, MedChem Express) staining. Cells were seeded in 24-well plates and treated as indicated in each experiment. Subsequently, 200 μL PBS containing 5 μg/mL PI was added to each well and images were captured by using a fluorescence microscope.

## Flow cytometry analysis
Cells were seeded in 12-well plates. After indicated treatments in each experiment, cells were digested with trypsin and harvested by centrifugation at 1000 x g for 5 min. The cell pellets were resuspended in HBSS or PBS containing corresponding dye, then incubated in the dark at 37 °C for 20 min. The cells were washed with ice-cold PBS twice, resuspended in 200 μL PBS, and analyzed by using the indicated laser of a flow cytometer (CytoFLEX LX, Beckman) for excitation. At least 10000 cells were acquired for each sample from three independent experiments. Flow cytometry was analyzed with *FlowJo* v10.4.0. The relative MFI (mean fluorescence intensity) was calculated and shown.

## Lipid peroxide measurement
For flow cytometry analysis of cellular lipid peroxides, cells were seeded in 12-well plates. After indicated treatments in each experiment, cells were digested and resuspended in HBSS supplemented with 2 μM BODIPY 581/591-C11 dye (D3861, Thermo Fisher Scientific) in the dark at 37 °C for 20 min. After washed with PBS twice, cells were analyzed by using the 488-nm laser of a flow cytometer (CytoFLEX LX, Beckman) for excitation. For confocal microscopy imaging of lipid peroxides, cells were seeded in confocal dishes with glass bottom. After indicated treatments, cells were washed with PBS twice and incubated with working solution containing 2 μM BODIPY 581/591-C11 dye in the dark at 37 °C for 20 min. After washed with PBS, images were captured by using a N-STORM microscope (Nikon, Japan).

## ROS measurement

Cells were seeded in 12-well plates. After indicated treatments, cells were harvested by trypsin digestion and resuspended in PBS supplemented with 5 μM DCFH-DA (D399, Thermo Fisher Scientific) in the dark at 37 °C for 20 min. After washed with PBS twice, cells were resuspended in PBS and analyzed by flow cytometry. At least 10000 cells were counted in each sample from three independent experiments. The relative MFI was calculated and shown.

## Lysosomal localization of PMO-DiI

Cells were seeded in confocal dishes with glass bottom and grown overnight. After treated with PMO-DiI for 4 h, cells were incubated in culture medium containing 50 nM LysoTracker-Green (C1047S, Beyotime Biotechnology) for 15 min at 37°C. Subsequently, cells were washed with PBS twice, and imaged by using a N-STORM microscope (Nikon, Japan).

## Lysosomal lipid peroxidation analysis

Cells were seeded in confocal dishes and incubated for 24 h. After indicated treatments, cells were incubated with working solution containing 1.0 μM Foam-LPO[30] and 50 nM LysoTracker-Blue (L7525, Thermo Fisher Scientific) for 15 min at 37 °C. After washed with PBS twice, cells were imaged with a N-STORM microscope (Nikon, Japan).

## Intracellular Fe$^{2+}$ determination

For confocal microscopy imaging, cells were cultured in confocal dishes with glass bottom and incubated for 24 h. After indicated treatments, cells were stained with 1 μM FerroOrange (F374, Dojindo), with or without 50 nM LysoTracker-Green, for 20 min at 37 °C. After washed with PBS, cells were imaged by using a N-STORM microscope (Nikon, Japan). For flow cytometry analysis of intracellular Fe$^{2+}$, cells were seeded in 12-well plates and treated as indicated in each experiment. Cells were digested and incubated with PBS containing 1 μM FerroOrange in the dark for 20 min at 37°C. After washed with PBS, cells were analyzed by using the 543-nm laser of a flow cytometer (CytoFLEX LX, Beckman) for excitation.

## Immunofluorescence staining

For cultured cells, cells were seeded in a 24-well plate preplaced with a sterilized round coverslip. After indicated treatments in each experiment, cells were fixed with 4% paraformaldehyde for 15 min at 37 °C, then stored in PBS in 4 °C for following immunofluorescence staining. For tissue slides, samples were soaked with xylene and ethanol to remove paraffin, then boiled in citrate buffer (10 mM sodium citrate, pH 6.0) for 10 min for antigen retrieval. Subsequently, both fixed cell and tissue slides were permeabilized with 0.1% Triton X-100 for 10 min at room temperature, blocked with 5% goat serum for 1 h at room temperature, stained with the specific primary antibody at 4 °C overnight, and washed three times with PBS, successively. Next, samples were stained with corresponding fluorescent secondary antibodies. After washed with PBS three times, the fixed cell and tissue slides were stained with DAPI (C1002, Beyotime Biotechnology) and mounted in antifade mountant. Images were captured by using a N-STORM microscope (Nikon, Japan). The fluorescence intensity was analyzed by using *ImageJ* 1.8.0 software.

## Transfection

Indicated cells were seeded in a 24-well plate preplaced with a sterilized round coverslip in each well. After grown overnight, cells were transfected with Lipo8000™ transfection reagent (C0533, Beyotime Biotechnology) according to the manufacturer's instruction. Briefly, 1 μg plasmid and 0.8 μL Lipo8000™ transfection reagent were gently mixed in 100 μL Opti-MEM. After stewing for 20 min at room temperature, the mixture was dropwisely added into culture medium and

the plate was incubated for 6 h. Then, the culture medium was changed with fresh DMEM supplemented with 10% FBS.

## GSH assay

Cells were seeded in 6-well plates and cultured overnight. After indicated treatments, cells were harvested and cell numbers were determined. The identical number of cells ($2 \times 10^6$ cells for each sample) were taken for further GSH measurement by using a commercial GSH kit (S0052, Beyotime Biotechnology) according to the manufacturer's instruction.

## Western blot

After indicated treatments, cells in 6-well plates were scraped in NP40 lysis buffer (P0013, Beyotime Biotechnology) supplemented with phosphatase and protease inhibitor cocktail, then lysed for 10 min on ice. The lysate was isolated through centrifugation at 12000 x g at 4 °C for 10 min. After determination of protein concentration, the lysate was added with proper volume of 5 × loading buffer, then boiled for 15 min. Samples were resolved on SDS-PAGE gels and transferred to a nitrocellulose membrane. The nitrocellulose membrane was blocked with 5% non-fat milk or 5% BSA at room temperature for 1 h, then incubated with indicated primary antibody overnight at 4 °C. After washed with TBST (Tris buffered saline with Tween-20, which is formulated as 137 mM NaCl, 20 mM Tris, 0.1% Tween-20, pH 7.6) three times, the nitrocellulose membrane was incubated with HRP-conjugated secondary antibody for 1 h at room temperature, and the protein bands were visualized by adding chemiluminescence reagents. The band density was analyzed by *ImageJ* 1.8.0 software. The relative expressions of each protein normalized to the corresponding internal reference protein were quantitatively calculated and shown in the histograms. The uncropped scans of the blots are provided in the Source Data file.

## Real-time PCR

Total RNA was extracted by using Trizol reagent (4992730, Tiangen Biotech) according to the manufacture's instruction. After determination of RNA concentration, 1 μg RNA was reverse transcribed to cDNA with an ABScript III RT Master Mix (RK20429, ABclonal). Real-time PCR was conducted in a 10-μL reaction mixture by using Universal SYBR Green Fast qPCR Mix (RK21203, ABclonal). Subsequently, the PCR program was run as follows: 95°C, 3 min; 39 cycles (95°C, 5 s; 60 °C, 30 s for each cycle). The primers used in this study were listed in Supplementary Table 2.

## APAP-induced liver injury

APAP-induced liver injury and PMO intervention in mice was conducted as shown in Fig. 7a. Briefly, APAP (HY-66005, MedChem Express) was dissolved in sterilized PBS at 10 mg/mL. Male C57BL/6 J mice (8-week-old, 18–20 g) were intravenously injected with PMO at 20 mg/kg bodyweight 24 h before APAP intoxication. The control mice were injected with sterilized PBS with the same volume. These mice were fasted for 12 h but given free access to water. Subsequently, the mice were intravenously injected with PMO at 20 mg/kg bodyweight or sterilized PBS once again, and intraperitoneally injected with APAP at 200 mg/kg bodyweight. After another 24 h, the mice were sacrificed and tissues were isolated for further analysis.

## Liver ischemia/reperfusion injury

Liver ischemia/reperfusion injury and PMO intervention in mice was conducted as shown in Fig. 8a. Briefly, male C57BL/6 J mice (8-week-old, 18–20 g) were intravenously injected with PMO at 20 mg/kg bodyweight 24 h before liver ischemia/reperfusion. The control mice were injected with sterilized PBS with the same volume. These mice were fasted for 12 h but given free access to water. Subsequently, the mice were intravenously injected with PMO at 20 mg/kg bodyweight or

sterilized PBS once again, and then anesthetized with inhaled iso-flurane. An atraumatic clip was used to block hepatic artery and hepatic portal vein to drive ischemia for 60 min, and then the atrau-matic clip was removed to initiate hepatic reperfusion for another 12 h. The mice were sacrificed and tissues were isolated for further analysis.

## ICP-MS

ICP-MS was carried out to measure Mn concentrations after PMO supplementation. For those samples of cell lysate and culture med-ium, cells were seeded in 55-cm² dishes and incubated with PMO for 4 h. Subsequently, the culture medium was collected. The cells were washed three times with PBS, and then harvested by trypsin diges-tion. After determination of cell number, 1 mL NP-40 was added to lyse the cell pellet with identical cell number. Both the culture medium and cell lysate were dialyzed in dialysis bags in deionized water. Dialysis was performed at room temperature for 24 h in 50-mL centrifuge tubes. The solution in the centrifuge tubes was collected for the following ICP-MS. For the tissue samples, 0.1 g tissue was weighed into a plastic tube containing 4 mL $HNO_3$ and 1 mL $HClO_4$. The tube was placed onto a microwave digestion instrument and maintained at 180°C for 3 h until digestion was complete. After cooling down to room temperature, the sample was diluted to 10 mL with a solution containing 2% $HNO_3$. Mn standard solutions with five different concentrations was prepared by gradient dilution, and all samples were automatically spiked with 200 ppb Ge as the internal standard. Mn concentrations were measured with an iCAP Q (Thermo Fisher). Both standard solutions and test solutions were measured for three times.

## Histological analysis and immunohistochemistry

Mice were sacrificed and tissues were rapidly isolated, fixed in 4% paraformaldehyde and embedded in paraffin. Embedded tissues were sectioned at a thicknesses of 5–10 μm. These tissue sections were deparaffinized and rehydrated, then stained with hematoxylin and eosin. Images were captured with a microscope. For immunohis-tochemistry, tissue sections were successively deparaffinized and rehydrated, incubated in 3% $H_2O_2$ solution in methanol at room tem-perature for 10 min to block endogenous peroxidase activity, and boiled in citrate buffer for 10 min for antigen retrieval. Subsequently, samples were stained with indicated primary antibody overnight at 4 °C and the corresponding secondary antibody for 30 min at room temperature. DAB substrate solution was added to reveal the color of antibody staining. The slides were mounted and images were captured by using a microscope. For histological analysis and immunohis-tochemistry assays, at least 6 mice were used for each group. For each sample, 5 independent views were photographed, each with an area of 0.3 mm².

## TUNEL staining

Liver tissue was fixed in 4% paraformaldehyde, embedded in paraffin and then serially sectioned. TUNEL staining was conducted by using a commercial kit (C1091, Beyotime Biotechnology) according to the manufacturer's instruction.

## Bioluminescence imaging

Male C57BL/6 J mice (8-week-old, 18–20 g) were intravenously injected with PMO-DiI at 20 mg/kg bodyweight. The mice were sacrificed 24 h postinjection. Major tissues, including the heart, liver, spleen, lung, and kidney, were isolated. The biodistribution of PMO-DiI in these tissues was visualized with an IVIS Spectrum Imaging System (Perki-nElmer, America).

## Serum biochemical test

Blood was collected and serum was obtained by centrifugation of the blood at 860 x g for 10 min at 4 °C. Serum ALT, AST, BUN, CR, CK-MB and LDH were measured by using an automatic biochemistry analyzer.

## MDA assay

Liver tissue was weighed and the same weight of tissue was homo-genized with equal volume of RIPA lysis buffer (P0013B, Beyotime Biotechnology). Protein concentration was measured by using a BCA kit (P0011, Beyotime Biotechnology). MDA concentration was mea-sured by using a commercial MDA kit (S0131S, Beyotime Biotechnol-ogy) according to the manufacturer's instruction, and the MDA level was normalized to the protein content.

## Liver iron measurement

Liver iron was measured by using a tissue iron assay kit (E-BC-K773-M, Elabscience Biotechnology) according to the manufacturer's instruction. Briefly, liver tissues were weighed and the same weight of tissue was homogenized with the solution buffer on ice. The lysates were collected and chromogenic solution was added. The solution was mixed and incubated at 37 °C for 10 min. Finally, the absorbance of each sample was measured with a microplate reader at 593 nm. The iron concentrations were calculated based on the standard solutions with gradient iron concentrations, and normalized to the tissue weight.

## Statistics and reproducibility

All experiments were repeated at least 3 times with similar results to ensure the reproducibility of results. Measurements were taken from distinct samples, and no sample was measured repeatedly. Graphs were drawn and statistics were analyzed by using *GraphPad Prism* 8 software. Two-tailed unpaired Student's t test was used to compare two groups and the data were expressed as mean ± SEM. $P < 0.05$ is statistically significant.

## Reporting summary

Further information on research design is available in the Nature Portfolio Reporting Summary linked to this article.

# Data availability

Source data are provided as a Source Data file. Source data are pro-vided in this paper.

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

## Acknowledgements

This work was supported by the National Natural Science Foundation of China (32070738 to H.W., 32071971 to H.H.W.), the National Key Research and Development Program of China (2022YFD2300205 to H.H.W.), the Fundamental Research Funds for the Central Universities (2662021JC001 to H.W.), Agricultural Microbiology of Large Research Infrastructures (463119009 to H.W.), and HZAU-AGIS Cooperation Fund (SZYJY2021008 to H.H.W.).

## Author contributions

H.W. and H.H.W. conceived and designed the research. X.Y.S., J.H.Li and J.H.Liu performed most of the experiments. B.L.F., T.Z., Q.L. and H.X.M. performed additional experiments. H.W. and H.H.W. co-wrote the paper.

## Competing interests

The authors declare no competing interests.
