## [Peer Review File · Nature Communications]

REVIEWER COMMENTS

Reviewer #1 (Remarks to the Author):

Comments to the author

In this manuscript, authors developed a simple one-step synthesis of ultra-small poly(acrylic) acid-coated Mn₃O₄ nanoparticles (PMO) with therapeutic activity against acute liver injury. In particular, PMO exhibits multi-enzymatic activity in scavenging ROS and could effectively suppress ferroptosis. The authors presented novel observations and offered advancements in the field. This report can be published after a minor revision addressing my concern

Key Findings:

1. PMO supplementation potentially neutralizes the ferroptosis induced by RSL-3, as confirmed by using a chemical inhibitor, Ferrostatin-1, and Erastatin (specific ferroptosis inhibitor).
2. Author proved that the cellular uptake mode of PMO is carried out through macropinocytosis and not by other forms of endocytosis. This experiment was supported by the use of three potential inhibitors amiloride (micropinocytosis), dynasore (caveolin-mediated endocytosis), and filipin (clathrin-mediated endocytosis).
3. The biomarkers like ALT, AST, BUN, LDH, etc., taken to prove that PMO reduces the severity of ferroptosis, are well presented in Figures 6-8.

Critique:-

1. Author says PMO nanoparticles perform high multiple enzymatic activities in Introduction para-5, line – 122. However, PMO nanoparticles showed less enzymatic activity (Catalase, SOD, and peroxidase) than other Mn₃O₄ nanoparticles reported earlier (Figure 1e-g).
2. Author should provide enough evidence to characterize the surface modification of Mn₃O₄ by polyacrylic acid by FT-IR or RAMAN spectroscopy or EDX.
3. Quantitative cytotoxicity/cell viability assay measurement will be appreciated by MTT assay/Alamar Blue assay/Flow cytometry rather than microscopy assay (line 164, Figure S1a).
4. After various times, the major (Please check the line and add a suitable word, like time points in line 312).
5. Scale bar looks inconsistent in microscopy images of figure S1.
6. Fig 1: The scheme requires a slight correction. The authors' data do not mention the direct chelation of Fe by PMO. The lowering of Fe load by PMO is indirect at best.
7. Fig 2 shows that PMO fails to restore the GSH balance upon Erastin exposure which might be due to the effect of Erastin on the transporter, which is not affected by PMO. However, the levels of GSH in PMO-RSL3 samples better justify the role of PMO in restoring the redox potential of the cell.
8. What is the half-life of PMO in the cell or its cellular retentivity?
9. Fig 4: The authors propose that PMO inhibits Ferritinophagy. Although the data with free reactive iron looks convincing, only minimal change is observed for FTH, which may be a consequence of high ROS. Further supportive evidence with additional Ferritinophagy determinants such as NCOA will be more appreciated.
10. The authors propose that PMO restores mTOR phosphorylation. Does PMO directly mediate

phosphor-transfer reactions? Since mTOR is a crucial metabolic inducer, the authors should hint at the reason behind maintaining the mTOR phosphorylation state. Also, the authors should comment on why GPX4 levels remain reduced (Fig2m,n) when PMO restores p-mTOR levels. mTOR is known to induce GPX4 transcription.

11. Fig6: The authors should mention the timelines for the organ toxicity analysis. Since PMO distributes primarily into the liver, is there any change in the liver histology after 16D treatment when the Mn levels normalize?

12. Fig7 and Fig8: Although FTH levels show minor variation in APAP and LIRI samples concerning PMO treated, does it reflect on the bioactive Fe levels in the tissues? Authors should consider quantifying the FTH band intensities in all the immunoblots presented.

Reviewer #2 (Remarks to the Author):

The manuscript by Xinyi Shan et al. entitled "Targeting Ferroptosis by Poly(acrylic) Acid Coated Mn3O4 Nanoparticles Alleviates Acute Liver injury" demonstrates for the first time that Mn3O4 nanoparticles can prevent ferroptosis in vitro and in vivo through their versatile antioxidant activities. After carefully reading the manuscript, I have multiple concerns and cannot recommend publication in the current state. In many cases, quantitative results are necessary to assess effects, while qualitative approaches were used (i.e. microscopy to demonstrate differences in cell viability or toxicity). The authors must address the following major points to consider re-submission.

Major points:

Major text and language editing required to improve readability and allow publication. Some examples: Normally, the past tense is used to present results.

Line 254: "determining ferroptosis vulnerability" ◊ This is wrong◊ maybe "determining the iron metabolism of cells."

Line372: delete: "hemorrhagic shock, severe sepsis and trauma" or add appropriate citations.

Line378: Sadly, there is currently no reliable way to determine ferroptosis levels in mice. Please add we measured end products of lipid peroxidation (4-HNE and MDA)

Line 419-Line438: missing text block?

Line 483: revise sentence?

Line586-617: Missing text block? FACS method?

Scheme 1: Fenton reaction is actually producing hydroxyl radicals from H₂O₂. Superoxide radicals are mostly produced as a by-product of the respiratory chain and detoxified by superoxide dismutase to H₂O₂. The scheme might be misleading regarding the contribution of the Fenton reaction.

In my opinion, the mechanism of PMO uptake and degradation would requires some more experimental support. I am suggesting to test the effect of BafA1 treatment (blocking of autophagosome-lysosome fusion) on the uptake, localization and functionality of PMOs. This experiment would add important information regarding the late stage of uptake and early stage of PMO degradation.

This being said, while micropinocytosis certainly has a large contribution in the uptake of PMOs, the data indicates that at least 40% of the uptake is dynamin-dependent (dynasore) (Figure 3e+f) with a smaller contribution of clathrin-dependent uptake. The text should reflect this better. Is the combined use of inhibitors feasible?

Cell viability assays are required to support the claims made by the others (i.e., Figure 2 a+c,

Figure 3h, Figure 5c-d, Figure S1 b-g, S6 a-b, S7 a+c).

Similarly, relative mean fluorescence intensities (MFI) are much more accurately and reliably determined using FACS analysis rather than microscopy pictures of a limited amount of events (As already done in Figure 2e-h). The authors should also apply this method to Figure 2k+l, Figure 3b+d, Figure 3f.

The claim that PMO-Dil are proximal to or in association with the actin network is for me "worthless", as everything is actually proximal or in association with actin. This claim does not add confidence that micropinocytosis involved.

Quantification of co-localization of markers i.e., PMO-Dil with LysoTrackerGreen (Figure 4a) as well as other markers used (Figure 4f-k) would add more confidence to the presented data.

Quantifications of p-mTOR/mTOR signal in figure 5a+b is required. A Torin positive control is necessary!

Please also include a RSL3/Erastin + Torin control in Fig. 5e + f

Side note, Torin has an IC₅₀ of 2-10 nM. In this study 5 μM were used, while 250nM were previously reported to completely inhibit cell proliferation (maybe cell type differences).

Question: Would To

ALT/AST measurements after liver IRI are required to assess the liver damage of the whole organ (add to Figure 8). This was already done for the APAP (see Figure 7 b+c).

How much total area was analysed to acquire the necrotic area (%) (Figure e+g+Figure 8 c+e). There was no statement how many animals were used. Please show single data points for all histograms (Fig. 7+ Fig.8).

How do you explain the increase in IL-1β level in the PMO control (Figure S8d)?

FACs method is missing.

Statistical statements regarding animal experiments are missing.

Uncropped blots need to be provided in order to assess the western blot results!

- What are the noteworthy results?

The application of ultrasmall poly(acrylic) acid coated Mn₃O₄ nanoparticles (PAA@Mn₃O₄-NPs, PMO)/ ROS scavengers to alleviate ferroptosis in vitro and in vivo (liver IRI).

- Will the work be of significance to the field and related fields? How does it compare to the established literature? If the work is not original, please provide relevant references.

Even though the redox activity of Mn₃O₄ nanoenzymes has been characterized in detail elsewhere (PMID: 28922532), their application to mitigate ferroptosis is new. Especially the strong protective effects of PMO during liver IRI are noteworthy. A preprint on ResearchSquare (<https://doi.org/10.21203/rs.3.rs-2288620/v1>) has similarly applied Mn₃O₄ nanozymes to treat myocardial ischemia reperfusion injury, but did not mention to specifically target ferroptosis with this treatment. While the possibility of coating Mn₃O₄ nanoparticles and other nanomaterials with biocompatible polyacrylic acid has been known for a long time (e.g. PMID: 21125087), the reported manufacturing process for these specific Mn₃O₄ nanoparticles and their in vivo applicability is, to my knowledge, new.

It is known that mTORC1 is activated by ROS, whereas H₂O₂ scavengers could prevent activation. In this context, PMOs act in a similar manner as an H₂O₂ and general ROS scavenger.

- Does the work support the conclusions and claims, or is additional evidence needed?

Additional experimental evidence is necessary.

- Are there any flaws in the data analysis, interpretation and conclusions? –Do these prohibit publication or require revision?

- Is the methodology sound? Does the work meet the expected standards in your field?

Microscopy is not the correct method to state quantitative differences in cell death research, cell viability assays or FACS analysis are required. Microscopy can be shown as supportive data. An appropriate cell viability assay (CCK8) with quantitative readout was already used in this manuscript. Similarly, quantitative cell death analysis using PI requires subsequent FACS analysis instead of qualitative microscopy readouts as used throughout the manuscript (i.e., Figure 2 a+c, Figure 3h, Figure 5c-d, Figure S1 b-g, S6 a-b, S7 a+c). Similarly, relative mean fluorescence intensities (MFI) are much more accurately and reliably determined using FACS analysis rather than microscopy pictures of a limited amount of events.

- Is there enough detail provided in the methods for the work to be reproduced?

FACs method is missing.

Statistical statements regarding animal experiments are missing.

Uncropped blots need to be provided in order to assess the westernblot results!

Reviewer #3 (Remarks to the Author):

The authors report the one-pot synthesis of biocompatible poly(acrylic) acid coated Mn₃O₄ nanoparticles (PAA@Mn₃O₄-NPs, PMO). PMO are able to alleviate acute liver injury by inhibiting ferroptosis. This research has been some novelty, with somewhat efficacious data. However, the English writing should be improved for better presentation and many concerns should be cleared by authors. The questions are listed below.

1. In the PMO characterization part, the PDI (Polydispersity Index) is missing

2. In Fig. 2/4/5/7/8, western blot experiments results should be quantitative and please mark the significance.

3. In Figure 7I, the internal reference protein band does not meet the specification, please repeat this experiment.

4. In the study of cytotoxicity experiment, the author used a microscope to observe the number of cells directly. This method is not representative and accurate enough. Please use MTT colorimetry or other methods.

5. In vitro experiments, the author successfully demonstrated the ability of PMO to inhibit ferroptosis in fibroblasts, but the compatibility between fibroblasts and liver injury model is not good. Please repeat the vitro experiment with hepatocytes.

6. To assess the target ability and ROS-responsive feature, the TME image of PMO in H₂O₂ solution and in vivo bioluminescence distribution with PMO should be evaluated.

7. In vivo experimental results show that PMO can alleviate acute liver injury, but the existing data can only prove that PMO plays a role by reducing oxidative stress. Please add more experiments related to ferroptosis pathway.

**REVIEWER COMMENTS**

Reviewer #1 (Remarks to the Author):

Comments to the author

In this manuscript, authors developed a simple one-step synthesis of ultra-small poly(acrylic)
acid-coated Mn₃O₄ nanoparticles (PMO) with therapeutics activity against acute liver injury. In
particular, PMO exhibits multi-enzymatic activity in scavenging ROS and could effectively
suppress ferroptosis. The authors presented novel observations and offered advancements in the
field. This report can be published after a minor revision addressing my concern

Key Findings:

1. PMO supplementation potentially neutralizes the ferroptosis induced by RSL-3, as confirmed
by using a chemical inhibitor, Ferrostatin-1, and Erastatin (specific ferroptosis inhibitor).

2. Author proved that the cellular uptake mode of PMO is carried out through macropinocytosis
and not by other forms of endocytosis. This experiment was supported by the use of three
potential inhibitors amiloride (micropinocytosis), dynasore (caveolin-mediated endocytosis), and
filipin (clathrin-mediated endocytosis).

3. The biomarkers like ALT, AST, BUN, LDH, etc., taken to prove that PMO reduces the severity
of ferroptosis, are well presented in Figures 6-8.

**Response:** The authors appreciate the reviewer's positive comments and stating that our study is
of novelty and offers advancements in the field. We have followed the suggestions, carried out
additional experiments and revised the manuscript accordingly. Thank you very much for the
critical reading and professional suggestions.

Critique:-

1. Author says PMO nanoparticles perform high multiple enzymatic activities in Introduction
para-5, line – 122. However, PMO nanoparticles showed less enzymatic activity (Catalase, SOD,
and peroxidase) than other Mn₃O₄ nanoparticles reported earlier (Figure 1e-g).

**Response:** Thank you for pointing out this and we compared the enzymatic activities between
PMO and the earlier reported Mn₃O₄ nanoparticles. Due to the different dosages, different
methodologies and different working time used in these studies, quantitative comparisons between
PMO and the earlier reported Mn₃O₄ nanoparticles are difficult. Generally, for the SOD-like O₂⁻
scavenging activity and OH• scavenging activity, PMO performed comparable or a litter lower
activities, when compared to the Mn₃O₄ nanoparticles¹, BSA-Mn₃O₄ NPs (bovine serum
albumin-modified Mn₃O₄ nanoparticles)² and Mn₃O₄ nanoflowers³. However, PMO displayed
lower Catalase-like H₂O₂ scavenging activity, when compared to Mn₃O₄ nanoparticles¹, Mn₃O₄
nanoparticles with flower-like morphology⁴, BSA-Mn₃O₄ nanoparticles² and dMn₃O₄
(1-dodecanethiol stabilized Mn₃O₄) nanoparticles⁵. We think that the high OH• scavenging
activity partially conferred the ferroptosis inhibitory capacity of PMO, as OH• is the most reactive
radical and the indeed ROS which directly attacks PUFAs for the generation of lipid peroxides
and initiates ferroptosis^{6,7}. Taken together, we agree with your comment and thus revise this claim
to “PMO nanoparticles could scavenge broad-spectrum ROS, especially OH•, thus could

efficiently reduce lipid peroxidation and counteract ferroptotic cell death.”, as presented in line
123-125 in this new manuscript.

2. Author should provide enough evidence to characterize the surface modification of Mn_3O_4 by
polyacrylic acid by FT-IR or RAMAN spectroscopy or EDX.

**Response:** According to this suggestion, we carried out FTIR analysis to further characterize the
surface modification of PMO. FTIR results confirmed the existence of -OH and C=O groups in
PMO, suggesting that PAA was successfully coated on Mn_3O_4 nanoparticles (Figure S1 in this
revised manuscript). We also added the corresponding text description in line 147-149 in this
revised manuscript.

3. Quantitative cytotoxicity/cell viability assay measurement will be appreciated by MTT
assay/Alamar Blue assay/Flow cytometry rather than microscopy assay (line 164, Figure S1a).

**Response:** We measured the relative cell viability to confirm the lower cytotoxicity of PMO by
using CCK-8 assay. The new data has been added in Figure S2a, and the corresponding
description has been added in line 167-168 in this revised manuscript. In addition to the original
microscopy imaging, these cell viability measurements further confirmed that PMO nanozymes
are cytocompatible.

We repeated all of the cytotoxicity experiments and measured the relative cell viability by using
CCK-8 assay. Specifically, for the Figure 2a and 2c, the corresponding CCK-8 assays were shown
below (Figure R1), as similar results have been presented in Figure 2b, 2d, as well as in Figure
S2b-S2e. The data suggested that PMO could effectively counteract ferroptosis (Figure R1). For
original Figure 3h, the corresponding CCK-8 assay was shown in the new Figure 3h. The data
suggested that co-treatment of Amiloride would restart RSL3-induced ferroptosis under PMO
supplementation. For Figure 5c and 5d, the corresponding CCK-8 assays were shown in the new
Figure 5c and 5d. The data suggested that Torin would partially restore RSL3- and
Erastin-induced ferroptosis under PMO supplementation. For original Figure S1b-g, the
corresponding CCK-8 assays were shown in the new Figure S2b-g. The data suggested that PMO
could effectively counteract ferroptosis in a dose-dependent manner, PMO provided a long-time
protection, and that PMO could effectively counteract ferroptosis in HT1080 cells. For original
Figure S3, The relative cell viability was added in the new Figure S4. The data suggested that
PMO failed to inhibit apoptosis induced by the anti-cancer drugs. For original Figure S4a and S4b,
The relative cell viability was added in the new Figure S5a and S5b. The data suggested that
PMO-DiI could effectively counteract ferroptosis. For original Figure S6a and S6b, the
corresponding CCK-8 assays were shown in the new Figure S7a and S7b. The data suggested that
the lysosomal antioxidant CYE could counteract ferroptosis. For the original Figure S7a, the
corresponding CCK-8 assays were shown in the new Figure S9a, S9b and S10a. The data
suggested that PMO effectively counteracted ferroptosis in mouse hepatocytes. For the original
Figure S7c, the corresponding CCK-8 assays were shown in the new Figure S10b-d. The data
suggested that PMO effectively counteracted ferroptosis in human hepatocytes. Accordingly, we
revised the corresponding Result section and Figure Legends section.

Figure R1: PMO effectively counteract ferroptosis (related to Figure 2a and 2c in the manuscript). MEF cells were treated with 1 μ M RSL3 (left) or 10 μ M Era (right) with or without 50 μ g/mL PMO or 10 μ M Fer. The relative cell viability was analyzed by CCK-8 assay. Data represent the mean \pm SEM from three independent samples.

Side note: This figure is also presented in line 372-402 and line 601-629 in this Response Letter, as the reviewer #2 and reviewer #3 raised the similar concern.

4. After various times, the major (Please check the line and add a suitable word, like time points in line 312).

Response: We are sorry for missing this important information. The exact time points (1 day, 14 days and 30 days post-injection) have been added in line 320-321 and line 330 in this revised manuscript.

5. Scale bar looks inconsistent in microscopy images of figure S1.

Response: Yes, the scale bars in Figure S1f and S1g (correspond to Figure S2f and S2g in this new version) are not consistent with those in other microscopy images in Figure S1a-S1e (correspond to Figure S2a-S2e in this new version), due to the different magnification. We cropped the Figure S2f and S2g to make all the scale bars in Figure S2 consistent. Please see the new Figure S2 in this revised manuscript.

6. Fig 1: The scheme requires a slight correction. The authors' data do not mention the direct chelation of Fe by PMO. The lowering of Fe load by PMO is indirect at best.

Response: The authors thank you for the critical reading. PMO reduced the intracellular labile iron through suppressing ferritinophagy-mediated iron release, but not through direct chelation of iron. To avoid the misunderstanding, we followed your suggestion and revised this scheme as shown in Scheme 1 in this revised manuscript.

7. Fig 2 shows that PMO fails to restore the GSH balance upon Erastin exposure which might be due to the effect of Erastin on the transporter, which is not affected by PMO. However, the levels of GSH in PMO-RSL3 samples better justify the role of PMO in restoring the redox potential of the cell.

Response: Erastin and RSL3 are two typical ferroptosis inducers. However, the underlying mechanism differs sharply. Specifically, Erastin targets the System Xc⁻ to suppress cystine uptake,

leading to reduced GSH synthesis, while RSL3 directly binds to GPX4 and disrupts the enzymatic
 activity of GPX4. Therefore, exposure of Erastin results in rapid exhaustion of GSH, while
 exposure of RSL3 cannot change GSH level due to GPX4 inactivation⁸. In this study, we
 measured the GSH levels in the Erastin-treated cells with or without PMO administration. The
 result in Figure 2o suggested that Erastin induced notable GSH reduction, consistent with the
 previous report⁸. PMO administration failed to restore the GSH level. This may be due to that
 PMO can not affect Erastin binding and suppressing System Xc⁻, just as you mentioned. In
 addition, we followed your suggestion and measured the GSH level in the RSL3-treated cells with
 or without PMO. The result below (Figure R2) showed that RSL3 exposure failed to change the
 intracellular GSH, which is consistent with the previous report⁸.

 **Figure R2: Both PMO and RSL3 fail to alter the intracellular GSH.** Cells were treated by RSL3 with or
 without PMO. The relative GSH level was analyzed. Data represent the mean \pm SEM from three independent
 samples.

8. What is the half-life of PMO in the cell or its cellular retentivity?

**Response:** To characterize the cellular retentivity of PMO, we incubated cells with PMO-DiI, and
 then measured the fluorescence during several rounds of cell passage in 5 days. The data presented
 below showed that the mean fluorescence intensity gradually decreased with cell passages (Figure
 R3). This was greatly due to the cell proliferation. The total fluorescence intensity was slightly
 reduced (Figure R3). In addition, the biocorrosion assay through measuring the possible Mn
 release after PMO administration by ICP-MS further confirmed the biostability of PMO (Figure
 3i).

**Figure R3: The cellular retentivity of PMO.** Cells were incubated with PMO-DiI, and the cellular fluorescence

intensities were monitored during several rounds of cell passage in 5 days. The representative histogram of flow
cytometry analysis was shown in (a). The relative MFI and relative cell number, both normalized to those in first
148 day, were shown in (b). Data represent the mean \pm SEM from three independent samples.

9. Fig 4: The authors propose that PMO inhibits Ferritinophagy. Although the data with free
reactive iron looks convincing, only minimal change is observed for FTH, which may be a
consequence of high ROS. Further supportive evidence with additional Ferritinophagy
determinants such as NCOA will be more appreciated.

**Response:** Our published paper⁹ and other papers^{10,11} have reported that ferroptosis inducers could
initiate ferritinophagy, which mediates ferritin degradation, leading to iron release and expediting
ferroptotic cell death. However, ferritin is always increased, but not decreased, upon the exposure
of ferroptosis inducers, as shown in Figure 4b, 4c, and Figure S9l, S9m. This could be explained
by that ferritinophagy-mediated iron release would transcriptionally activate ferritin expression^{10,11},
as shown by the qRT-PCR assay in Figure 4d, 4e, and Figure S9n, S9o. Therefore, the
coinstantaneous ferritin degradation and ferritin synthesis may lead to minimal change in FTH.
However, the analysis of intracellular bioactive iron (Figure 2i-2l), lysosomal translocation of
ferritin (Figure 4f, 4g) and lysosomal iron (Figure 4h, 4i), collectively demonstrated that PMO
inhibited ferritinophagy.

In this revised manuscript, we followed this suggestion and blotted NCOA4, the receptor protein
which mediates selective ferritinophagy. The new data presented in Figure 4b and 4c showed that
both Erastin and RSL3 led to NCOA4 decrease, which could be suppressed by PMO
administration. We have added this description in line 263 and line 269 in this revised manuscript.
In addition, we also blotted NCOA4 in RSL3- and Erastin-treated hepatocytes. The data shown in
Figure S9l and S9m further suggested that PMO administration mitigated NCOA4 degradation
during ferroptosis induction.

10. The authors propose that PMO restores mTOR phosphorylation. Does PMO directly mediate
phosphor-transfer reactions? Since mTOR is a crucial metabolic inducer, the authors should hint at
the reason behind maintaining the mTOR phosphorylation state.

Also, the authors should comment on why GPX4 levels remain reduced (Fig2m,n) when PMO
restores p-mTOR levels. mTOR is known to induce GPX4 transcription.

**Response:** Firstly, we followed this suggestion and carried out a phosphor-transfer reaction to
check whether PMO could directly activate mTOR. We transfected cells with flag-mTOR, then
the cells were treated by HBSS for 1 h to induce mTOR dephosphorylation or maintained in the
full medium as a control. The flag-mTOR recombinant protein was affinity-purified through
immunoprecipitation by using anti-flag antibody. The flag-mTOR protein was incubated with
PMO in the kinase reaction buffer containing ATP for 1 h, and then mTOR phosphorylation was
checked by Western blot (Figure R4a). The result below showed that PMO failed to elevate
mTOR phosphorylation in a cell-free system (Figure R4b). Moreover, the PMO administration
alone gave rise to minimal effect on mTOR phosphorylation, when compared to the control group
(Figure 5a, 5b, and Figure R4c). PMO exposure also failed to preserve mTOR phosphorylation in
cells treated by mTOR inhibitor Torin (Figure 5a and 5b, and Figure R4c) or HBSS (Figure R4c).
Collectively, these data indicate that PMO can not directly activate mTOR (at least in a cell-free

system), but only preserve mTOR activation under pro-ferroptotic state.

It has been well understood that mTOR is dynamically activated in the lysosomal surface. Dozens
or more proteins have been identified to manipulate mTOR activation, making the underlying
mechanism of mTOR activation very complicated and far from thoroughly understood^{12,13}. To be
honest, currently we have no idea on the exact mechanism for PMO preserving mTOR activation
under pro-ferroptotic state. It has been widely reported that excessive ROS would lead to mTOR
inactivation¹⁴⁻¹⁶. In this context, a reasonable possibility may exist that the lysosome-resident
PMO could scavenge lysosomal ROS and protect mTOR from inactivation by ROS during
ferroptosis induction. We have added this hypothesis in line 492-494.

As you mentioned, mTOR is known to facilitate GPX4 transcription¹⁷. However, GPX4 protein is
manipulated at several different aspects. Besides transcriptional modulation by mTOR-4EBP
signaling axis¹⁷ and by several transcriptional factors¹⁸, it has been gradually recognized that
GPX4 undergoes degradation under pro-ferroptotic stimuli. Wu and colleagues reported that
Erastin promotes GPX4 degradation in a CMA (chaperone-mediated autophagy)-dependent
manner¹⁹. GPX4 could also be degraded through selective macroautophagy, and TAX1BP1 (Tax1
binding protein 1) functions as the receptor protein²⁰. Furthermore, it was reported that
ubiquitin-proteasome system is also involved in GPX4 degradation. TRIM46 might be one of the
E3 ligases to mediate GPX4 ubiquitination²¹. In this study, we found that PMO failed to preserve
GPX4 during ferroptosis induction (Figure 2m, 2n, and Figure S9e, S9f). This is possible due to
that PMO administration could not change these mechanisms for GPX4 degradation.

**Figure R4: PMO preserve mTOR activity not through the direct phosphor-transfer reaction.** (a) Schematic
illustration of phosphor-transfer reaction and PMO intervention. Briefly, cells were transfected with flag-mTOR,
treated by HBSS for 1 h to induce mTOR dephosphorylation or left untreated as a control. The flag-mTOR was
purified through immunoprecipitation by using anti-flag antibody. The flag-mTOR protein was incubated with
PMO in the kinase reaction buffer (25 mM Tris pH 7.5, 5 mM β -glycerol-phosphate, 2 mM DTT, 0.1 mM Na_3VO_4 ,
10 mM MgCl_2 , 2 mM ATP) for 1 h. (b) mTOR phosphorylation of samples in (a) was checked by Western blot. (c)
Cells were treated by HBSS or Torin for 1 h, with or without PMO. mTOR phosphorylation was checked by
Western blot.

11. Fig6: The authors should mention the timelines for the organ toxicity analysis. Since PMO
distributes primarily into the liver, is there any change in the liver histology after 16D treatment

when the Mn levels normalize?

**Response:** We are sorry for missing this important information. As you can see in line 320-321 in
the Result section, as well as in line 906-907 in the Figure Legends section, we added the detailed
timelines (1 day, 14 days and 30 days post-injection) for this organ toxicity analysis. After 14 days
of PMO injection, the liver performed normal histology, which has been shown in Figure 6a. We
also supplemented a new H&E staining of liver tissue isolated from mice after 16 days of PMO
injection, according to this suggestion. The data presented below (Figure R5) showed a normal
liver histology. Therefore, we think that PMO are biocompatible.

**Figure R5: Intravenously injected PMO fail to change the liver histology.** C57BL/6J mice were intravenously
injected with PMO at 20 mg/kg bodyweight. After 16 days of injection, liver was isolated and H&E staining was
conducted to show the histological morphology. Scale bar=20 μ m.

12. Fig7 and Fig8: Although FTH levels show minor variation in APAP and LIRI samples
concerning PMO treated, does it reflect on the bioactive Fe levels in the tissues? Authors should
consider quantifying the FTH band intensities in all the immunoblots presented.

**Response:** Firstly, we followed this suggestion and measured tissue iron in liver isolated from
mice in the two liver injury models. The data presented in Figure 7l and Figure 8l showed that
both APAP exposure and LIRI elevated liver iron, which were similar to several previous
reports²²⁻²⁶. More importantly, PMO injection could significantly reduced this iron deposition,
further confirming that PMO mitigate ferroptosis in these two animal models of acute liver injury.
Secondly, we measured the band density of FTH blots in Figure 7m and Figure 8m by using
*ImageJ* software, and then the relative expressions of FTH to Actin were quantitatively analyzed.
The data were presented in the right histograms in Figure 7m and Figure 8m in this revised
manuscript. Moreover, we also quantified other blots and the data have been added in Figure 2m,
2n, 4b, 4c, 5a, 5b, 7n, 8n, S7c, S7d, S9e, S9f, S9l, S9m, S9r and S9s in this revised manuscript.

Thank you very much for your critical reading and for all professional suggestions.

Reviewer #2 (Remarks to the Author):

The manuscript by Xinyi Shan et al. entitled “Targeting Ferroptosis by Poly(acrylic) Acid Coated
Mn₃O₄ Nanoparticles Alleviates Acute Liver injury” demonstrates for the first time that Mn₃O₄
nanoparticles can prevent ferroptosis in vitro and in vivo through their versatile antioxidant
activities. After carefully reading the manuscript, I have multiple concerns and cannot recommend
publication in the current state. In many cases, quantitative results are necessary to assess effects,

while qualitative approaches were used (i.e. microscopy to demonstrate differences in cell
viability or toxicity). The authors must address the following major points to consider
re-submission.

**Response:** The authors would like to thank you for the critical reading and for the professional
suggestions. We followed these suggestions and carried out additional experiments to further
confirm our claims. Specifically, we repeated all of the cytotoxicity and cell viability experiments
by using CCK-8 assay. Then the relative cell viability was calculated and added in this revised
manuscript. We also utilized flow cytometry to quantify the intracellular bioactive iron and the
PMO-DiI uptake, which were analyzed through microscopy imaging in the original manuscript.
Please see our point-by-point response below.

Major points:

Major text and language editing required to improve readability and allow publication. Some
examples: Normally, the past tense is used to present results.

**Response:** We have invited a professional English editor to polish the language. Specifically, we
have changed the tense to the past tense when describing the results. All of the revisions have been
highlighted in blue in this revised manuscript.

Line 254: “determining ferroptosis vulnerability” ◊ This is wrong◊ maybe “determining the iron
metabolism of cells.”

**Response:** We changed “determining ferroptosis vulnerability” to “determining cellular iron
metabolism” in line 260 (correspond to line 254 in the original manuscript).

Line 372: delete: “hemorrhagic shock, severe sepsis and trauma” or add appropriate citations.

**Response:** We followed this suggestion and deleted “hemorrhagic shock, severe sepsis and
trauma” in line 397 (correspond to line 372 in the original manuscript).

Line 378: Sadly, there is currently no reliable way to determine ferroptosis levels in mice. Please
add we measured end products of lipid peroxidation (4-HNE and MDA)

**Response:** We changed this sentence “In addition, we measured the ferroptosis levels in these
mice.” to “In addition, we evaluated the ferroptosis levels in these mice by measuring the end
products of lipid peroxidation, including 4-HNE and MDA.” in line 403-405 (correspond to line
378 in the original manuscript).

Line 419-Line 438: missing text block?

**Response:** We checked the PDF file, and there is no problem in line 419-438 in the original
manuscript (correspond to line 448-468 in this revised manuscript).

Line 483: revise sentence?

**Response:** We revised this sentence to “Additionally, liver ischemia/reperfusion injury is a major
risk factor for liver transplantation, which is the only effective therapy for end-stage liver diseases
including hepatic viral infection, alcohol abuse, and non-alcoholic steatohepatitis.” in line 512-515
(correspond to line 482-484 in the original manuscript).

Line 586-617: Missing text block? FACS method?

**Response:** We checked the line 586-617 in the original PDF file (correspond to line 625-654 in
this revised manuscript), there is no missing text block in this part. Actually, we described the
FACS method in the “Lipid peroxide measurement” section in line 585-595 in the original
manuscript. In this revised manuscript, we emphasized this method and introduced FACS
separately in the “Flow cytometry analysis” section in line 615-622. To avoid duplicate
description, we also revised the “Lipid peroxide measurement” line 624-632 section accordingly.

Scheme 1: Fenton reaction is actually producing hydroxyl radicals from H_2O_2 . Superoxide radicals
are mostly produced as a by-product of the respiratory chain and detoxified by superoxide
dismutase to H_2O_2 . The scheme might be misleading regarding the contribution of the Fenton
reaction.

**Response:** The authors appreciate this professional suggestion. We revised the scheme and
emphasized that PMO suppress ferritinophagy-mediated iron release, decelerate iron-catalyzed
Fenton reaction, and thus reduce the production of OH^\bullet , which is the exact ROS directly attacking
PUFAs to generate lipid peroxidation. PMO could also directly scavenge OH^\bullet , $O_2^{\bullet-}$ and H_2O_2 ,
while the latter two could expedite lipid peroxidation by converting to OH^\bullet . Furthermore, PMO
preserve mTOR activity. Through these multifaceted mechanisms, PMO perform potent
anti-ferroptosis activity, and could alleviate ferroptosis-associated liver damage, including
APAP-induced and hepatic ischemia/reperfusion-induced liver injury. Please see the revised
Scheme 1 in this revised manuscript.

In my opinion, the mechanism of PMO uptake and degradation would requires some more
experimental support. I am suggesting to test the effect of BafA1 treatment (blocking of
autophagosome-lysosome fusion) on the uptake, localization and functionality of PMOs. This
experiment would add important information regarding the late stage of uptake and early stage of
PMO degradation.

**Response:** We followed this suggestion and checked the uptake, localization and functionality of
PMO upon the administration of chloroquine, a widely-used autophagy inhibitor that blocks
autophagosome-lysosome fusion²⁷. The flow cytometry analysis showed that co-treatment of
chloroquine failed to change the cellular uptake of PMO-DiI (Figure R6a and R6b). The confocal
microscopy imaging showed that PMO-DiI colocalized with lysosomal marker protein LAMP2,
which was not disturbed by chloroquine administration (Figure R6c). Furthermore, we also tested
whether inhibition of autophagosome-lysosome fusion could alter the ferroptosis inhibition of
PMO. The cell viability measurement by CCK-8 assay showed that chloroquine gave rise to
negligible effect on this ferroptosis inhibition of PMO (Figure R6d and R6e). Collectively, these
data suggest that blocking autophagosome-lysosome fusion fails to change the uptake, localization
and functionality of PMO.

 **Figure R6: Blocking autophagosome-lysosome fusion by chloroquine would not change the uptake,**
 **localization and functionality of PMO.** (a) Cells were treated with PMO-DiI with or without chloroquine (CQ)
 for the indicated time, and flow cytometry was carried out to measure the intracellular fluorescence intensities. (b)
 The relative mean fluorescence intensities were calculated and shown in the histogram. (c) Cells were incubated
 with 20 $\mu\text{g}/\text{mL}$ PMO-DiI with or without CQ. Then the cells were fixed and stained with anti-LAMP2 antibody.
 Images were captured to show the co-localization between PMO-DiI and LAMP2. Scale bar=10 μm . (d, e) Cells
 were treated with RSL3 (d) or Erastin (e), with or without PMO and CQ. The relative cell viability was analyzed
 by CCK-8 assay. Data represent the mean \pm SEM from three independent samples.

 This being said, while micropinocytosis certainly has a large contribution in the uptake of PMOs,
 the data indicates that at least 40% of the uptake is dynamin-dependent (dynasore) (Figure 3e+ f)
 with a smaller contribution of clathrin-dependent uptake. The text should reflect this better. Is the
 combined use of inhibitors feasible?

**Response:** According to this suggestion, we revised our claim in line 233-235 in the Result
 section, as well as line 455-458 in the Discussion section, to emphasize that micropinocytosis and
 clathrin-mediated endocytosis, but not caveolin-mediated endocytosis, contribute to PMO uptake.
 Additionally, we used the two inhibitors, Amiloride and Dynasore, to treat cells together, and
 found that combined suppression of these two pathways led to an almost complete inhibition of
 PMO-DiI uptake (Figure R7), further suggesting that micropinocytosis and clathrin-mediated
 endocytosis mediate PMO uptake synergistically.

 **Figure R7: PMO uptake is dependent on micropinocytosis and clathrin-mediated endocytosis.** MEF cells

were incubated with PMO-DiI with or without Amiloride and Dynasore. The relative fluorescence intensity was
analyzed by flow cytometry analysis, and shown in the right histogram. Data represent the mean \pm SEM from
three independent samples.

Cell viability assays are required to support the claims made by the others (i.e., Figure 2 a+c,
Figure 3h, Figure 5c-d, Figure S1 b-g, S6 a-b, S7 a+c).

**Response:** We repeated these cytotoxicity experiments and measured the relative cell viability by
using CCK-8 assay. Specifically, for Figure 2a and 2c, the corresponding CCK-8 assays were
shown below (Figure R8), as similar results have been presented in Figure 2b, 2d, as well as in
Figure S2b-S2e. The data suggested that PMO could effectively counteract ferroptosis (Figure R8).
For original Figure 3h, the corresponding CCK-8 assay was shown in the new Figure 3h. The data
suggested that co-treatment of Amiloride would restart RSL3-induced ferroptosis under PMO
supplementation. For Figure 5c and 5d, the corresponding CCK-8 assays were shown in the new
Figure 5c and 5d. The data suggested that Torin would partially restore RSL3- and
Erastin-induced ferroptosis under PMO supplementation. For original Figure S1b-g, the
corresponding CCK-8 assays were shown in the new Figure S2b-g. The data suggested that PMO
could effectively counteract ferroptosis in a dose-dependent manner, PMO provided a long-time
protection, and that PMO could effectively counteract ferroptosis in HT1080 cells. For original
Figure S6a and S6b, the corresponding CCK-8 assays were shown in the new Figure S7a and S7b.
The data suggested that the lysosomal antioxidant CYE could counteract ferroptosis. For the
original Figure S7a, the corresponding CCK-8 assays were shown in the new Figure S9a, S9b and
S10a. The data suggested that PMO effectively counteracted ferroptosis in mouse hepatocytes. For
the original Figure S7c, the corresponding CCK-8 assays were shown in the new Figure S10b-d.
The data suggested that PMO effectively counteracted ferroptosis in human hepatocytes.
Accordingly, we revised the corresponding Result section and Figure Legends section.

In addition, we also repeated the cytotoxicity experiments in the original Figure S1a, original
Figure S3, as well as original Figure S4a and S4b, by using CCK-8 assay. The relative cell
viability has been added in the new Figure S2a, Figure S4, as well as Figure S5a and S5b in this
revised manuscript. The data suggested that PMO nanozymes were cytocompatible (Figure S2a),
PMO failed to inhibit apoptosis induced by the anti-cancer drugs (Figure S4), and that PMO-DiI
could effectively counteract ferroptosis (Figure S5a and S5b).

**Figure R8: PMO effectively counteract ferroptosis** (related to Figure 2a and 2c in the manuscript). MEF cells

were treated with 1 μ M RSL3 (left) or 10 μ M Era (right) with or without 50 μ g/mL PMO or 10 μ M Fer. The
relative cell viability was analyzed by CCK-8 assay. Data represent the mean \pm SEM from three independent
samples.

**Side note:** This figure is also presented in line 58-91 and line 601-629 in this Response Letter, as the reviewer #1
and reviewer #3 raised the similar concern.

Similarly, relative mean fluorescence intensities (MFI) are much more accurately and reliably
determined using FACS analysis rather than microscopy pictures of a limited amount of events
(As already done in Figure 2e-h). The authors should also apply this method to Figure 2k+l,
Figure 3b+d, Figure 3f.

**Response:** We followed this suggestion and calculated the relative mean fluorescence intensities
(MFI) by using flow cytometry analysis. Specifically, the intracellular bioactive iron was
measured by FerroOrange staining followed by flow cytometry analysis. The representative
histogram of flow cytometry analysis and the relative MFI were shown in the new Figure 2i and 2j.
The data further confirmed that PMO reduced the cellular labile iron under pro-ferroptotic state.
We also revised the description in line 200-201 in the Result section, and line 844-846 in the
Figure Legends section. The relative MFI shown in Figure 3b and 3d in the original manuscript
were calculated based on flow cytometry analysis, and we have added the representative
histogram of the flow cytometry analysis in the new Figure 3b and 3d and emphasized this point
in line 228-229 in the Result section, as well as line 856-857 and line 860-861 in the Figure
Legends section. In addition, we monitored the PMO-DiI uptake by using flow cytometry analysis.
The representative histogram of flow cytometry analysis and relative MFI was shown in the new
Figure 3f. The data further suggested that micropinocytosis and clathrin-mediated endocytosis, but
not caveolin-mediated endocytosis, contributed to PMO uptake. We revised the description in line
863-865 in the Figure Legends section in this revised manuscript.

The claim that PMO-DiI are proximal to or in association with the actin network is for me
“worthless”, as everything is actually proximal or in association with actin. This claim does not
add confidence that micropinocytosis involved.

**Response:** The authors thank you for this constructive suggestion, and we deleted this claim and
the corresponding data in the original Figure 3g.

Quantification of co-localization of markers i.e., PMO-DiI with LysoTrackerGreen (Figure 4a) as
well as other markers used (Figure 4f-k) would add more confidence to the presented data.

**Response:** We quantitatively analyzed these co-localization by using *ImageJ*, and the statistical
data were shown in Figure S6 in this revised manuscript. Specifically, Pearson's correlation
coefficient showed a good co-localization between LysoTracker-Green and PMO-DiI in Figure 4a
(Figure S6a). The relative fluorescence intensities of lysosomal FTL (Figure 4e, 4f), lysosomal
FerroOrange (Figure 4g, 4h) and lysosomal Foma-LPO (Figure 4i, 4j) were quantitatively
analyzed and shown in Figure S6e-S6j. The data further confirmed that exposure of RSL3 and
Erastin led to increased lysosomal translocation of ferritin, lysosomal iron and lysosomal lipid
peroxidation, and these could be mitigated by PMO administration (Figure S6e-S6j). In addition,
we also quantitatively analyzed the co-localizations between PMO-DiI and the markers of
lysosome, ER or mitochondrion in the new Figure S6b-S6d, and the Pearson's correlation

coefficient showed that PMO-DiI had a good co-localization with lysosome, but not
mitochondrion or ER (Figure S6b-S6d).

Quantifications of p-mTor/mTor signal in figure 5a+b is required. A Torin positive control is
necessary!

**Response:** Firstly, a Torin control has been added in the new Figure 5a and 5b in this revised
manuscript. The data showed that Torin treatment as a positive control could overwhelmingly
reduce mTOR activity. PMO administration could preserve mTOR phosphorylation upon
pro-ferroptosis insults, but failed to maintain mTOR activity under Torin treatment. We also
added the corresponding text description in line 304-305 in the Result section and line 893-895 in
the Figure Legends section. Secondly, we measured the band densities of the p-mTOR and mTOR
blots in Figure 5a and 5b by using the *ImageJ* software, and then the relative p-mTOR normalized
to mTOR was quantitatively calculated and shown in Figure 5a and 5b.

Please also include a RSL3/Erastin + Torin control in Fig. 5e + f

**Response:** We followed this suggestion and included a RSL3+Torin control in Figure 5e and a
Erastin+Torin control in Figure 5f. Torin administration gave rise to minimal effect on RSL3- and
Erastin-induced lipid peroxidation. This was consistent with the cell viability measurement in
Figure 5c and 5d, and could be explained by that exposure of RSL3 or Erastin alone could trigger
mTOR inactivation. We also changed the corresponding text description in Line 307-310 in this
revised manuscript.

Side note, Torin has an IC50 of 2-10 nM. In this study 5 μ M were used, while 250 nM were
previously reported to completely inhibit cell proliferation (maybe cell type differences).

**Response:** When Torin was used to inhibit mTOR activity, we followed some published papers.
Zhou et, al. and Crosby et, al. used Torin at 1 μ M^{28,29}. Ma et, al. used Torin at 10 μ M³⁰. Moreover,
we observed no abnormality when used 5 μ M Torin. 5 μ M Torin didn't inhibit cell proliferation or
induce cell death, which was suggested by the cell viability analysis in Figure 5c and 5d.

Question: Would To

ALT/AST measurements after liver IRI are required to assess the liver damage of the whole
organ (add to Figure 8). This was already done for the APAP (see Figure 7 b+c).

**Response:** We repeated this mice experiment and measured the serum ALT and AST. The data
were presented in the new Figure 8b and 8c in this revised manuscript. The data showed that PMO
injection could significantly decrease serum ALT and AST in mice with LIRI, further suggesting
that PMO could alleviate LIRI associated liver injury. We also added the corresponding text
description in line 400-401 in the Result section, and line 934-935 in Figure Legends section.

How much total area was analysed to acquire the necrotic area (%) (Figure e+g+Figure 8 c+e).
There was no statement how many animals were used. Please show single data points for all
histograms (Fig. 7+ Fig.8).

**Response:** For H&E staining in Figure 7e and Figure 8e (correspond to Figure 8c in the original
manuscript), 9 mice were used for each group. For TUNEL staining in Figure 7g and Figure 8g
(correspond to Figure 8e in the original manuscript), 6 mice were used for each group. For each

mice, we photographed 5 independent views, each with an area of 0.3 mm². The necrotic area
analysis was confirmed by two different people independently. We added this important
information in line 766-768 in Materials and methods section. In addition, we followed this
suggestion and showed single data points in all histograms in Figure 7 and Figure 8 in this revised
manuscript.

How do you explain the increase in IL-1 β level in the PMO control (Figure S8d)?

**Response:** In the original data shown in Figure S8d, the relative mRNA level of IL-1 β increased
slightly in the PMO group. However, no statistical significance existed between the control group
and PMO group. This may be due to the individual variation and relative lower expression of
IL-1 β in liver. To confirm this, we repeated this real time PCR by using more samples, and the
data shown in the new Figure S11d in this revised manuscript suggested no difference in the IL-1 β
mRNA between the control group and PMO group.

FACs method is missing.

Statistical statements regarding animal experiments are missing.

Uncropped blots need to be provided in order to assess the westernblot results!

**Response:** In this revised manuscript, we introduced FACS method in the “Flow cytometry
analysis” section in line 615-622 in the Materials and methods section. The statistical statements
regarding animal experiments have been added in line 766-768 in the Materials and methods
section. We also showed single data points in all histograms to show the independent mice
replicates used in Figure 7 and Figure 8. Moreover, all of the uncropped blots have been uploaded
in the supplementary Excel file entitled “Source Data”.

- What are the noteworthy results?

The application of ultrasmall poly(acrylic) acid coated Mn₃O₄ nanoparticles (PAA@Mn₃O₄-NPs,
PMO)/ ROS scavengers to alleviate ferroptosis in vitro and in vivo (liver IRI).

- Will the work be of significance to the field and related fields? How does it compare to the
established literature? If the work is not original, please provide relevant references.

Even though the redox activity of Mn₃O₄ nanoenzymes has been characterized in detail elsewhere
(PMID: 28922532), their application to mitigate ferroptosis is new. Especially the strong
protective effects of PMO during liver IRI are noteworthy. A preprint on ResearchSquare
(<https://doi.org/10.21203/rs.3.rs-2288620/v1>) has similarly applied Mn₃O₄ nanozymes to treat
myocardial ischemia reperfusion injury, but did not mention to specifically target ferroptosis with
this treatment. While the possibility of coating Mn₃O₄ nanoparticles and other nanomaterials with
biocompatible polyacrylic acid has been known for a long time (e.g. PMID: 21125087), the
reported manufacturing process for these specific Mn₃O₄ nanoparticles and their in vivo
applicability is, to my knowledge, new.

It is known that mTORC1 is activated by ROS, whereas H₂O₂ scavengers could prevent activation.

In this context, PMOs act in a similar manner as an H₂O₂ and general ROS scavenger.

**Response:** The authors are grateful to your positive comments and stating that our study is of
novelty in several aspects, including the anti-ferropotsis capacity, protective effects during acute
liver injury and easy manufacturing of PMO. Thank you!

- Does the work support the conclusions and claims, or is additional evidence needed?

Additional experimental evidence is necessary.

**Response:** We followed all of the suggestions and carried out additional experiments to verify our
claims. Please see the detailed information in the point-by-point response above.

- Are there any flaws in the data analysis, interpretation and conclusions? –Do these prohibit
publication or require revision?

- Is the methodology sound? Does the work meet the expected standards in your field?

Microscopy is not the correct method to state quantitative differences in cell death research, cell
viability assays or FACS analysis are required. Microscopy can be shown as supportive data. An
appropriate cell viability assay (CCK8) with quantitative readout was already used in this
manuscript. Similarly, quantitative cell death analysis using PI requires subsequent FACS analysis
instead of qualitative microscopy readouts as used throughout the manuscript (i.e., Figure 2 a+c,
Figure 3h, Figure 5c-d, Figure S1 b-g, S6 a-b,S7 a+c). Similarly, relative mean fluorescence
intensities (MFI) are much more accurately and reliably determined using FACS analysis rather
than microscopy pictures of a limited amount of events.

**Response:** We followed this suggestion and repeated all of the cell viability and cytotoxicity
experiments by using CCK-8 assay. The relative cell viability was calculated and shown in this
revised manuscript (please refer to line 372-402 in this Response Letter). Moreover, we also
analyzed the relative mean fluorescence intensities by using FACS analysis, and the data have
been shown in the new Figure 2i, 2j, 3b, 3d and 3f in this revised manuscript (please refer to line
410-424 in this Response Letter).

- Is there enough detail provided in the methods for the work to be reproduced?

FACS method is missing.

Statistical statements regarding animal experiments are missing.

Uncropped blots need to be provided in order to assess the western blot results!

**Response:** In this revised manuscript, we introduced FACS method in the “Flow cytometry
analysis” section in line 615-622 in the Materials and methods section. The statistical statements
regarding animal experiments have been added in line 766-768 in the Materials and methods
section. We also showed single data points in all histograms to show the independent mice
replicates used in Figure 7 and Figure 8. Moreover, all of the uncropped blots have been uploaded
in the supplementary Excel file entitled “Source Data”.

Thank you very much for your critical reading and for all professional suggestions.

Reviewer #3 (Remarks to the Author):

The authors report the one-pot synthesis of biocompatible poly(acrylic) acid coated Mn₃O₄
nanoparticles (PAA@Mn₃O₄-NPs, PMO). PMO are able to alleviate acute liver injury by
inhibiting ferroptosis. This research has been some novelty, with somewhat efficacious data.
However, the English writing should be improved for better presentation and many concerns
should be cleared by authors. The questions are listed below.

**Response:** The authors are grateful to the reviewer’s positive comment that our study is of some

novelty and the data are efficacious. We have invited a professional English editor to polish our
English writing. In addition, we followed your suggestions and carried out additional experiments
to address the concerns. Thank you very much for the critical reading and for all professional
suggestions.

1. In the PMO characterization part, the PDI (Polydispersity Index) is missing

**Response:** The PDI value was added in the Figure 1c in this revised manuscript. Furthermore, we
also added the corresponding description in line 146 in this revised manuscript.

2. In Fig. 2/4/5/7/8, western blot experiments results should be quantitative and please mark the
significance.

**Response:** We measured the band densities by using the *ImageJ* software. The relative
expressions of each protein to the corresponding internal reference protein were quantitatively
analyzed and shown in the corresponding histograms in this revised manuscript. The statistical
significance was marked. Besides, we have also quantitatively analyzed the western blots in
Figure S7 and S9.

3. In Figure 7l, the internal reference protein band does not meet the specification, please repeat
this experiment.

**Response:** We repeated this blotting and added a new Actin blot in Figure 7n (correspond to
Figure 7l in the original version).

4. In the study of cytotoxicity experiment, the author used a microscope to observe the number of
cells directly. This method is not representative and accurate enough. Please use MTT colorimetry
or other methods.

**Response:** We repeated all of the cytotoxicity experiments and measured the relative cell viability
by using CCK-8 assay. Specifically, for the Figure 2a and 2c, the corresponding CCK-8 assays
were shown below (Figure R9), as similar results have been presented in Figure 2b, 2d, as well as
in Figure S2b-S2e. The data suggested that PMO could effectively counteract ferroptosis (Figure
R9). For original Figure 3h, the corresponding CCK-8 assay was shown in the new Figure 3h. The
data suggested that co-treatment of Amiloride would restart RSL3-induced ferroptosis under PMO
supplementation. For Figure 5c and 5d, the corresponding CCK-8 assays were shown in the new
Figure 5c and 5d. The data suggested that Torin would partially restore RSL3- and
Erastin-induced ferroptosis under PMO supplementation. For original Figure S1a, the
corresponding CCK-8 assay was shown in the new Figure S2a. The data suggested that PMO
nanozymes were cytocompatible. For original Figure S1b-g, the corresponding CCK-8 assays
were shown in the new Figure S2b-g. The data suggested that PMO could effectively counteract
ferroptosis in a dose-dependent manner, PMO provided a long-time protection, and that PMO
could effectively counteract ferroptosis in HT1080 cells. For original Figure S3, The relative cell
viability was added in the new Figure S4. The data suggested that PMO failed to inhibit apoptosis
induced by the anti-cancer drugs. For original Figure S4a and S4b, The relative cell viability was
added in the new Figure S5a and S5b. The data suggested that PMO-DiI could effectively
counteract ferroptosis. For original Figure S6a and S6b, the corresponding CCK-8 assays were
shown in the new Figure S7a and S7b. The data suggested that the lysosomal antioxidant CYE

could counteract ferroptosis. For the original Figure S7a, the corresponding CCK-8 assays were
shown in the new Figure S9a, S9b and S10a. The data suggested that PMO effectively
counteracted ferroptosis in mouse hepatocytes. For the original Figure S7c, the corresponding
CCK-8 assays were shown in the new Figure S10b-d. The data suggested that PMO effectively
counteracted ferroptosis in human hepatocytes. Accordingly, we revised the corresponding Result
section and Figure Legends section.

**Figure R9: PMO effectively counteract ferroptosis** (related to Figure 2a and 2c in the manuscript). MEF cells
were treated with 1 μM RSL3 (left) or 10 μM Era (right) with or without 50 μg/mL PMO. The relative cell
viability was analyzed by CCK-8 assay. Data represent the mean \pm SEM from three independent experiments.

**Side note:** This figure is also presented in line 58-91 and line 372-402 in this Response Letter, as the reviewer #1
and reviewer #2 raised the similar concern.

5. In vitro experiments, the author successfully demonstrated the ability of PMO to inhibit
ferroptosis in fibroblasts, but the compatibility between fibroblasts and liver injury model is not
good. Please repeat the vitro experiment with hepatocytes.

**Response:** Mouse embryonic fibroblast (MEF) is derived from 13.5-day-old embryos after
removing head and viscera. MEF is a model cell, which has been widely used in toxicological
assessments, biochemical analysis and functional characterization in biology study, due to its
functional relevance, ease of collection and rapid growth kinetics³¹. In this study, we used MEF
cells to demonstrate the ferroptosis-inhibitory capacity and the underlying mechanism of PMO.

To strengthen the physiological compatibility of the *in vitro* experiments, we followed this
suggestion and repeated the major findings by using mouse and human hepatocytes. Cell viability
assay showed that PMO administration potently inhibited RSL3- and Erastin-induced ferroptosis
in AML12 mouse hepatocytes (Figure S9a, b). Furthermore, PMO administration also mitigated
APAP-induced ferroptosis in AML12 (Figure S10a), as well as RSL3-, Erastin- and APAP-induced
ferroptosis in L02 human hepatocytes (Figure S10b-d). PMO also suppressed lipid peroxidation,
which was elevated by RSL3 and Erastin in mouse hepatocytes (Figure S9c, d). Likewise, PMO
administration failed to change the expression of GPX4 and FSP1 (Figure S9e, f). Bioactive iron
measurement suggested that PMO decreased the labile iron pool in mouse hepatocytes (Figure
S9g, h). PMO-DiI entered mouse hepatocytes in a time- and dose-dependent manner (Figure S9i,
j), and the intracellular PMO-DiI mainly co-localized with lysosome, as shown by the confocal
microscopy imaging (Figure S9k). In addition, we also observed that PMO could suppress
ferritinophagy in mouse hepatocytes, as evidenced by the reduced protein and mRNA levels of

FTH, as well as the decelerated NCOA4 degradation under pro-ferroptotic state (Figure S9l-o). In
this context, PMO administration potentially reduced lipid peroxidation in lysosome, as shown by
the co-staining of LysoTracker and Foma-LPO, the specific dye to label lysosomal lipid
peroxidation (Figure S9p, q). Moreover, PMO could also preserve mTOR phosphorylation under
pro-ferroptotic state in mouse hepatocytes (Figure S9r, s). Collectively, these data suggest that
PMO potentially counteract ferroptosis in hepatocytes, and the mechanisms are similar to those in
MEF cells.

We have added Figure S9 and Figure S10 to show the new data. In addition, the corresponding
description has been added in line 376-392 in this revised manuscript.

6. To assess the target ability and ROS-responsive feature, the TEM image of PMO in H₂O₂
solution and in vivo bioluminescence distribution with PMO should be evaluated.

**Response:** Firstly, the TEM imaging of PMO in H₂O₂ solution was carried out and the data were
presented below (Figure R10). The results showed that PMO were stable in H₂O₂ solution, with no
obvious difference of nanoparticles' properties including morphology and dispersibility between
PMO and PMO + H₂O₂. In addition, we also carried out the bioluminescence imaging of PMO-DiI
nanozymes. The data was added in Figure S8 in this revised manuscript. Collaborated with the
ICP-MS analysis (Figure 6c) and fluorescence imaging (Figure 6d), the bioluminescence imaging
further confirmed that the injected PMO were specifically enriched in liver.

**Figure R10: TEM image of PMO in H₂O₂ solution.** The white boxes were magnified in right. Scale bar=200 nm.

7. In vivo experimental results show that PMO can alleviate acute liver injury, but the existing
data can only prove that PMO plays a role by reducing oxidative stress. Please add more
experiments related to ferroptosis pathway.

**Response:** Lipid peroxidation is the fundamental biochemical hallmark of ferroptosis. Currently
there is no reliable method to directly determine lipid peroxidation in mice. 4-HNE and MDA are
two aldehyde secondary products of lipid peroxides, and thus are widely used as the biomarkers of
lipid peroxidation, but not just the markers of general oxidative stress. We measured 4-HNE and
MDA, and the data shown in Figure 7h, 7i and Figure 8h, 8i, suggested that PMO injection
reduced lipid peroxidation during APAP-induced liver injury and LIRI-associated liver injury. To
further confirm the ferroptosis inhibitory capacity of PMO *in vivo*, we followed this suggestion
and carried out additional experiments. Firstly, we measured the liver iron in mice in these two

models. The data presented in Figure 7l and Figure 8l showed that both APAP exposure and LIRI
elevated hepatic iron, which were similar to some previous reports²²⁻²⁶. PMO injection could
significantly reduce this hepatic iron deposition. Secondly, we checked the mRNA expression of
*Ptgs2*, which is regarded as a marker of ferroptosis³². The data presented in Figure 7k and Figure
8k showed that both APAP exposure and LIRI elevated *Ptgs2* mRNA, which could be notably
suppressed by PMO injection. In summary, these data further confirmed that PMO injection
suppressed ferroptosis and alleviated liver injury induced by APAP and LIRI.

Thank you very much for your critical reading and for all professional suggestions.

References

- Yao, J. *et al.* ROS scavenging Mn(3)O(4) nanozymes for in vivo anti-inflammation. *Chem Sci* **9**,
2927-2933, doi:10.1039/c7sc05476a (2018).
- Guo, X. *et al.* The protective effect of biom mineralized BSA-Mn(3)O(4) nanoparticles on HUVECs
investigated by atomic force microscopy. *Analyst* **147**, 2097-2105, doi:10.1039/d2an00483f (2022).
- Meng, L. *et al.* Reactive Oxygen Species- and Cell-Free DNA-Scavenging Mn(3)O(4) Nanozymes for
Acute Kidney Injury Therapy. *ACS Appl Mater Interfaces* **14**, 50649-50663,
doi:10.1021/acsami.2c16305 (2022).
- Singh, N., Savanur, M. A., Srivastava, S., D'Silva, P. & Mugesh, G. A Redox Modulatory Mn(3) O(4)
Nanozyme with Multi-Enzyme Activity Provides Efficient Cytoprotection to Human Cells in a
Parkinson's Disease Model. *Angew Chem Int Ed Engl* **56**, 14267-14271, doi:10.1002/anie.201708573
(2017).
- Choi, H. S. *et al.* Inflammation-sensing catalase-mimicking nanozymes alleviate acute kidney injury via
reversing local oxidative stress. *J Nanobiotechnology* **20**, 205, doi:10.1186/s12951-022-01410-z (2022).
- Frenette, M. & Scaiano, J. C. Evidence for hydroxyl radical generation during lipid (linoleate)
peroxidation. *J Am Chem Soc* **130**, 9634-9635, doi:10.1021/ja801858e (2008).
- Yin, H., Xu, L. & Porter, N. A. Free radical lipid peroxidation: mechanisms and analysis. *Chem Rev* **111**,
5944-5972, doi:10.1021/cr200084z (2011).
- Dixon, S. J. *et al.* Ferroptosis: an iron-dependent form of nonapoptotic cell death. *Cell* **149**, 1060-1072,
doi:10.1016/j.cell.2012.03.042 (2012).
- Wu, H., Liu, Q., Shan, X., Gao, W. & Chen, Q. ATM orchestrates ferritinophagy and ferroptosis by
phosphorylating NCOA4. *Autophagy*, 1-16, doi:10.1080/15548627.2023.2170960 (2023).
- Hou, W. *et al.* Autophagy promotes ferroptosis by degradation of ferritin. *Autophagy* **12**, 1425-1428,
doi:10.1080/15548627.2016.1187366 (2016).
- Gao, M. *et al.* Ferroptosis is an autophagic cell death process. *Cell Res* **26**, 1021-1032,
doi:10.1038/cr.2016.95 (2016).
- Saxton, R. A. & Sabatini, D. M. mTOR Signaling in Growth, Metabolism, and Disease. *Cell* **168**,
960-976, doi:10.1016/j.cell.2017.02.004 (2017).
- Liu, G. Y. & Sabatini, D. M. mTOR at the nexus of nutrition, growth, ageing and disease. *Nat Rev Mol*
*Cell Bio* **21**, 183-203, doi:10.1038/s41580-019-0199-y (2020).
- Reiling, J. H. & Sabatini, D. M. Stress and mTOR signaling. *Oncogene* **25**, 6373-6383,
doi:10.1038/sj.onc.1209889 (2006).
- Kim, D. H. *et al.* mTOR interacts with raptor to form a nutrient-sensitive complex that signals to the cell
growth machinery. *Cell* **110**, 163-175, doi:10.1016/s0092-8674(02)00808-5 (2002).

Chen, L. *et al.* Hydrogen peroxide inhibits mTOR signaling by activation of AMPK α leading to
apoptosis of neuronal cells. *Lab Invest* **90**, 762-773, doi:10.1038/labinvest.2010.36 (2010).

Zhang, Y. *et al.* mTORC1 couples cyst(e)ine availability with GPX4 protein synthesis and ferroptosis
regulation. *Nat Commun* **12**, 1589, doi:10.1038/s41467-021-21841-w (2021).

Dai, C. *et al.* Transcription factors in ferroptotic cell death. *Cancer Gene Ther* **27**, 645-656,
doi:10.1038/s41417-020-0170-2 (2020).

Wu, Z. *et al.* Chaperone-mediated autophagy is involved in the execution of ferroptosis. *Proc Natl Acad*
*Sci U S A* **116**, 2996-3005, doi:10.1073/pnas.1819728116 (2019).

Xue, Q. *et al.* Copper-dependent autophagic degradation of GPX4 drives ferroptosis. *Autophagy*, 1-15,
doi:10.1080/15548627.2023.2165323 (2023).

Zhang, J., Qiu, Q., Wang, H., Chen, C. & Luo, D. TRIM46 contributes to high glucose-induced
ferroptosis and cell growth inhibition in human retinal capillary endothelial cells by facilitating GPX4
ubiquitination. *Exp Cell Res* **407**, 112800, doi:10.1016/j.yexcr.2021.112800 (2021).

Wang, C. *et al.* Ulinastatin protects against acetaminophen-induced liver injury by alleviating ferroptosis
via the SIRT1/NRF2/HO-1 pathway. *Am J Transl Res* **13**, 6031-6042 (2021).

Li, H. *et al.* Kaempferol prevents acetaminophen-induced liver injury by suppressing hepatocyte
ferroptosis via Nrf2 pathway activation. *Food Funct* **14**, 1884-1896, doi:10.1039/d2fo02716j (2023).

Cai, X. *et al.* Astaxanthin Activated the Nrf2/HO-1 Pathway to Enhance Autophagy and Inhibit
Ferroptosis, Ameliorating Acetaminophen-Induced Liver Injury. *ACS Appl Mater Interfaces* **14**,
42887-42903, doi:10.1021/acscami.2c10506 (2022).

Wu, S. *et al.* Macrophage extracellular traps aggravate iron overload-related liver ischaemia/reperfusion
injury. *Br J Pharmacol* **178**, 3783-3796, doi:10.1111/bph.15518 (2021).

Li, Y. *et al.* Ischemia-induced ACSL4 activation contributes to ferroptosis-mediated tissue injury in
intestinal ischemia/reperfusion. *Cell Death Differ* **26**, 2284-2299, doi:10.1038/s41418-019-0299-4
(2019).

Mauthe, M. *et al.* Chloroquine inhibits autophagic flux by decreasing autophagosome-lysosome fusion.
*Autophagy* **14**, 1435-1455, doi:10.1080/15548627.2018.1474314 (2018).

Zhou, J. *et al.* Activation of lysosomal function in the course of autophagy via mTORC1 suppression and
autophagosome-lysosome fusion. *Cell Res* **23**, 508-523, doi:10.1038/cr.2013.11 (2013).

Crosby, P. *et al.* Insulin/IGF-1 Drives PERIOD Synthesis to Entrain Circadian Rhythms with Feeding
Time. *Cell* **177**, 896-909 e820, doi:10.1016/j.cell.2019.02.017 (2019).

30 Ma, S. *et al.* PDGF-D-PDGFR β signaling enhances IL-15-mediated human natural killer cell survival.
*Proc Natl Acad Sci U S A* **119**, doi:10.1073/pnas.2114134119 (2022).

Hansen, J. M. & Piorczynski, T. B. Use of Primary Mouse Embryonic Fibroblasts in Developmental
Toxicity Assessments. *Methods Mol Biol* **1965**, 7-17, doi:10.1007/978-1-4939-9182-2_2 (2019).

Yang, W. S. *et al.* Regulation of ferroptotic cancer cell death by GPX4. *Cell* **156**, 317-331,
doi:10.1016/j.cell.2013.12.010 (2014).

REVIEWER COMMENTS

Reviewer #1 (Remarks to the Author):

My technical concerns are addressed comprehensively in the revised manuscript by the authors. The manuscript's syntax-related issues need to be fixed before it is ready for publication.

Reviewer #2 (Remarks to the Author):

The manuscript has significantly improved. Some critical points remain to be addressed before publication in my opinion:

- Line 155: For me it looks like that PMO is acting more as an antioxidant itself, than acting like an antioxidant enzyme (recovering antioxidants). Maybe the wording should be adjusted to reflect this subtle difference in the manuscript.
- Line 158: Which enzyme in your opinion scavenges the hydroxyl radical directly? To my knowledge there is none, but radical trapping antioxidants can scavenge the radical.
- Line 234: Dynasore is a polyphenolic compound, which are known radical trapping antioxidants and ferroptosis inhibitors. This argues that the missing sensitization could be due to the direct anti-ferroptotic effect of dynasore! Please add a statement in the discussion or maintext.
- Histological images in Figure 7f+h and Figure 8d+f are inconsistent in staining, illumination and white balance settings. Also image quality is low. Evaluation is therefore difficult:
- Especially the APAP+PMO pictures in 7f show a rather green than blue nuclear staining compared to the other pictures presented.
- Similar inconsistency in 7h control vs PMO (very different staining pattern!). Additionally APAP+PMO in 7h looks like necrotic areas just more faint.
- F8d LIRI appears much darker than the others and in general the illumination is too dark.
- F8f LIRI picture white balance is completely different than the other pictures of the panel.

Reviewer #3 (Remarks to the Author):

Thank you for submitting your manuscript to our journal. I have carefully reviewed the revised version of your manuscript, and I have a few recommendations for further improvement before the manuscript can be accepted for publication.

Overall, the revised manuscript has addressed many of the issues raised in my previous review, and I am pleased to see that you have made a significant effort to improve the clarity and quality of the manuscript.

However, I do recommend that you further refine the Discussion section. While the content is generally sound, there are some areas where the language could be more concise and focused. For example, some sentences and paragraphs could be combined or rephrased to improve readability. I would also suggest that you more explicitly articulate the implications of your findings for future research or practical applications in this section. This will help readers understand the significance of your work and how it contributes to the broader field.

In summary, I believe that these changes will help strengthen the manuscript and improve its impact. Once these modifications have been made, I would be happy to recommend it for publication in our journal. I look forward to seeing the final version of your manuscript.

REVIEWER COMMENTS

Reviewer #1 (Remarks to the Author):

My technical concerns are addressed comprehensively in the revised manuscript by the authors. The manuscript's syntax-related issues need to be fixed before it is ready for publication.

Response: The authors are appreciated for the reviewer's positive comment on the revision that we made. We have invited a native English editor (from American Journal Experts (AJE, <https://www.aje.cn/>), who provides high quality language editing services and is recommended by Nature Research Editing Service) to polish our manuscript, and the revised parts have been highlighted in blue. Thank you very much for the critical reading.

Reviewer #2 (Remarks to the Author):

The manuscript has significantly improved. Some critical points remain to be addressed before publication in my opinion:

Response: The authors are appreciated for the reviewer's positive comment on the revision that we made. According to these new concerns, we carried out additional experiments and the data are listed as follows. Thank you very much for the critical reading.

- Line 155: For me it looks like that PMO is acting more as an antioxidant itself, than acting like an antioxidant enzyme (recovering antioxidants). Maybe the wording should be adjusted to reflect this subtle difference in the manuscript.

Response: Nanozymes are a large number of engineering nanomaterials with intrinsic enzyme-like activities [1]. In 2007, Yan and her colleagues verified the inherent peroxidase-like activity of Fe₃O₄ nanoparticles [2]. Since then, increasing number of nanomaterials with enzyme-mimicking activities have been synthesized and reported. So far, there are more than 2700 nanozyme-related papers published when “nanozyme” is retrieved as the key word in PubMed.

However, we do acknowledge that there are still some controversial opinions regarding the term of “nanozyme” [3, 4]. In terms of the Mn_3O_4 nanoparticles in this study, previous studies have defined them as nanozymes [5, 6]. In the current study, we found that PMO exhibited CAT-like H_2O_2 scavenging activity, SOD-like $\text{O}_2^{\cdot-}$ scavenging activity, as well as OH^{\cdot} scavenging activity. To further suggest that PMO act like antioxidant enzymes, but not just an antioxidant, additional experiments were conducted. Firstly, PMO were exposed to H_2O_2 solution or Fenton reaction solution (FeSO_4 and H_2O_2 to generate OH^{\cdot}) for 24 h, then PMO were isolated and subsequently used for H_2O_2 scavenging activity analysis and OH^{\cdot} scavenging activity analysis. The results showed that long exposure of PMO to H_2O_2 or Fenton reaction solution (to generate OH^{\cdot}) does not change the catalytic activity to detoxify ROS (Figure R1a, b). Secondly, TEM imaging of PMO incubated with H_2O_2 solution (for 24 h) was carried out, and the results showed that PMO were stable in H_2O_2 solution, with no obvious difference in the nanoparticles’ properties including morphology and dispersibility between PMO and PMO + H_2O_2 (Figure R1c). Thirdly, the DFT (Density Functional Theory) calculation showed that PMO can scavenge H_2O_2 and OH^{\cdot} . More importantly, PMO-mediated ROS scavenging is exothermic and can occur automatically (Figure R1d). Collectively, these results further suggest that PMO are antioxidant enzyme-mimicking nanozymes (or recovering antioxidants as you mentioned), but not just an antioxidant.

We do realize that some controversial opinions still exist regarding the use of “nanozyme”. Thus, we changed the wording and used “nanoparticles” when describing PMO, according to this suggestion.

Figure R1. PMO are antioxidant enzyme-mimicking nanozymes. (a, b) PMO were exposed to Fenton reaction solution (FeSO_4 and H_2O_2 to generate OH^{\cdot}) or H_2O_2 solution for 24 h. Then PMO were isolated by dialysis and (at the concentration of 25 $\mu\text{g}/\text{mL}$) used for OH^{\cdot} scavenging activity assay (a) or H_2O_2 scavenging activity assay (b), according to the methods reported in the “Materials and methods” in the manuscript. (c) TEM images of PMO in H_2O_2 solution. The white boxes were magnified in right. Scale bar=200 nm. (d) The Vienna Ab Initio Package (VASP) [7] was employed to perform all the spin-polarized density functional theory (DFT) calculations within the generalized gradient approximation (GGA) using the PBE formulation [8]. Side note: The TEM results in (c) were also presented in the Response Letter during the last revision, as the Reviewer #3 raised a similar concern.

- Line 158: Which enzyme in your opinion scavenges the hydroxyl radical directly? To my knowledge there is none, but radical trapping antioxidants can scavenge the radical.

Response: Yes, we agree with you that hydroxyl radical (OH•) can not be directly detoxified by any known antioxidant enzymes in biological systems [9], and we have introduced this point in line 131 and line 403 in the manuscript. In this study, ROS scavenging assays clearly showed that PMO exhibit high scavenging activity for OH•, which is the exact ROS directly triggering lipid peroxidation [10]. To our knowledge, there are several kinds of nanomaterials or nanozymes that perform OH• scavenging activity. In 2007, Singh and colleagues reported that Mn₃O₄ nanozymes (Mnf) functionally mimic antioxidant enzymes, including CAT (catalase), SOD (superoxide dismutase) and GPX (glutathione peroxidase). In addition, the Mn₃O₄ nanozymes could also scavenge OH• effectively [11]. This finding was further confirmed by a following study. Yao and colleagues reported that Mn₃O₄ nanozymes exhibit the antioxidant activity to detoxify OH• [5]. Furthermore, it has been widely reported that cerium oxide nanomaterials (nanoceria) with different size and distinct morphology could efficiently scavenge OH•, as shown in our published papers and others [12-16]. In Figure R1d, DFT (Density Functional Theory) calculation clearly showed that PMO can scavenge OH•. Importantly, PMO-mediated OH• scavenging is exothermic and can occur automatically (Figure R1d).

- Line 234: Dynasore is a polyphenolic compound, which are known radical trapping antioxidants and ferroptosis inhibitors. This argues that the missing sensitization could be due to the direct anti-ferroptotic effect of dynasore! Please add a statement in the discussion or maintext.

Response: Yes, Dynasore is indeed a polyphenolic compound, possesses radical trapping activity and thus a potential ferroptosis inhibitor [17]. By using PI staining and cell viability assay, we found that Dynasore could inhibit RSL3-induced ferroptosis at the concentration of 40 μM (Figure R2a, b), while 10 μM Dynasore was used to inhibit clathrin-mediated endocytosis (Figure 3e-h in the manuscript). Therefore, we think that Dynasore (10 μM) failing to restore RSL3-induced ferroptosis in the presence of PMO (Figure 3g, h in the manuscript) is due to that Dynasore exposure resulted in lower reduction in PMO uptake, but not due to that Dynasore directly inhibited ferroptosis (at least at the concentration of 10 μM).

To further suggest the potential involvement of clathrin-mediated endocytosis in PMO uptake, Pitstop2, another widely used inhibitor of clathrin-mediated endocytosis, was utilized. Pitstop2 exposure induces the inhibition of globular N-terminal β-propeller domain of clathrin to acutely interfere with clathrin-mediated endocytosis [18, 19]. We found that Pitstop2 exposure resulted in about 40% reduction in PMO-DiI uptake (Figure R2c), which is similar to that Dynasore works (Figure 3e, f in the manuscript). Cotreatment of Pitstop2 did not restore RSL3-induced ferroptosis in the presence of PMO, as shown by microscopy imaging and cell viability assay (Figure R2d, e).

Furthermore, we observed that exposure of Pitstop2 alone failed to inhibit ferroptosis induced by RSL3 (Figure R2f, g), suggesting that Pitstop2 itself does not possess anti-ferroptotic activity. Therefore, we concluded that PMO are internalized by cells mainly *via* macropinocytosis, while clathrin-mediated endocytosis may have lower contribution.

Figure R2. PMO are internalized by cells mainly *via* macropinocytosis. (a, b) Cells were treated with 1 μM RSL3, with or without ferroptosis inhibitor Fer-1 (ferrostatin-1) or the indicated dose of Dynasore. Cell death was visualized by PI staining (a). Scale bar = 50 μm. The relative cell viability was analyzed by CCK-8 assay (b). (c) Cells were pre-treated with Amiloride (1 mM) or Pitstop2 (15 μM) for 2 h, then incubated with 20 μg/mL PMO-DiI for 4 h. Fluorescence was analyzed by flow cytometry. Representative histograms of flow cytometry and the relative MFI were shown. (d, e) Cells were pre-treated with Amiloride (1 mM) or Pitstop2 (15 μM) for 2 h, then incubated with 1 μM RSL3 for 4 h, with or without 50 μg/mL PMO. Cell death was visualized by PI staining (d). Scale bar = 50 μm. The relative cell viability was measured by CCK-8 assay (e). (f, g) Cells were treated with 1 μM RSL3, with or without Fer-1 or the indicated dose of Pitstop2. Cell death was visualized by PI staining (f). Scale bar = 50 μm. The relative cell viability was analyzed by CCK-8 assay (g). For statistical analysis, data represent the mean ± SEM. n = 3 samples in b, c, e and g. ** $P < 0.01$, *** $P < 0.001$ was determined by student's t-test; ns, not significant.

- Histological images in Figure 7f+h and Figure 8d+f are inconsistent in staining, illumination and white balance settings. Also image quality is low. Evaluation is therefore difficult:

Response: We do notice that the quality of these images is low and there are some problems in the staining, illumination and white balance settings. Thus, we repeated these histological staining or re-captured the images, and the data have been shown in the new Figure 7f, 7h, 8d and 8f in this revised manuscript. Thank you for this kindly reminder.

- Especially the APAP+PMO pictures in 7f show a rather green than blue nuclear staining compared to the other pictures presented.

Response: These images are inconsistent in the nuclear staining, which is possibly due to the long-term deposit of the hematoxylin solution. We repeated these TUNEL staining and the data have been shown in the new Figure 7f. The results clearly showed that APAP exposure resulted in hepatocyte death, and this cell death could be largely mitigated by PMO injection.

- Similar inconsistency in 7h control vs PMO (very different staining pattern!). Additionally APAP+ PMO in 7h looks like necrotic areas just more faint.

Response: These images seem inconsistent due to the longer time of staining. We repeated these histological staining and the data have been shown in the new Figure 7h. The results clearly showed that APAP exposure resulted in elevated 4-HNE signal in the liver, and this increase in 4-HNE could be largely mitigated by PMO injection. Furthermore, there were relatively lower necrotic areas in the APAP+PMO sample, which was also suggested by the H&E staining in Figure 7d and TUNEL staining in Figure 7f.

- F8d LIRI appears much darker than the others and in general the illumination is too dark.

Response: We re-captured these images and the data have been shown in the new Figure 8d.

- F8f LIRI picture white balance is completely different than the other pictures of the panel.

Response: We do notice that these images are inconsistent in nuclear staining and in white balance settings. We repeated these TUNEL staining and the data have been shown in the new Figure 8f. The results clearly showed that LIRI resulted in notable cell death in the liver, and this cell death could be largely mitigated by PMO injection.

Thank you very much for your critical reading and for professional suggestions.

Reviewer #3 (Remarks to the Author):

Thank you for submitting your manuscript to our journal. I have carefully reviewed the revised version of your manuscript, and I have a few recommendations for further improvement before the manuscript can be accepted for publication.

Overall, the revised manuscript has addressed many of the issues raised in my previous review, and I am pleased to see that you have made a significant effort to improve the clarity and quality of the manuscript.

Response: The authors are appreciated for the reviewer's positive comment on the revision that we made. Thank you very much for your critical reading.

However, I do recommend that you further refine the Discussion section. While the content is generally sound, there are some areas where the language could be more concise and focused. For

example, some sentences and paragraphs could be combined or rephrased to improve readability. I would also suggest that you more explicitly articulate the implications of your findings for future research or practical applications in this section. This will help readers understand the significance of your work and how it contributes to the broader field.

In summary, I believe that these changes will help strengthen the manuscript and improve its impact. Once these modifications have been made, I would be happy to recommend it for publication in our journal. I look forward to seeing the final version of your manuscript.

Response: We revised the Discussion section according to this suggestion. Firstly, we deleted some sentences in the 1st paragraph to remove the redundant or repetitive phrasing. The 4th and 5th paragraphs in the previous Discussion were combined into one paragraph (line 425-442) to make the discussion of PMO working mechanisms more focused. Secondly, we rephrased the 6th paragraph in the previous Discussion (the 5th paragraphs in this new Discussion) to emphasize the clinical importance of biomedical research of APAP-induced acute liver injury and liver ischaemia/reperfusion injury, due to a shortage of effective treatments. We also added a new paragraph in line 463-468 and explicitly introduced the potential biomedical applications of PMO (PMO are promising candidates for clinical treatment of APAP-induced acute liver injury, and hold great promise for improving the clinical outcomes of liver surgeries, including liver resection and liver transplantation). Thirdly, we have invited a native English editor (from American Journal Experts (AJE, <https://www.aje.cn/>), who provides high quality language editing services and is recommended by Nature Research Editing Service) to polish our manuscript including the Discussion section. Thank you very much for your critical reading.

References

1. Jiang, D., et al., Nanozyme: new horizons for responsive biomedical applications. *Chem Soc Rev*, 2019. **48**(14): p. 3683-3704.
2. Gao, L., et al., Intrinsic peroxidase-like activity of ferromagnetic nanoparticles. *Nat Nanotechnol*, 2007. **2**(9): p. 577-83.

3. Scott, S., et al., Nano-Apples and Orange-Zymes. *Acs Catalysis*, 2020. **10**(23): p. 14315-14317.
4. Wei, H., et al., Nanozymes: A clear definition with fuzzy edges. *Nano Today*, 2021. **40**.
5. Yao, J., et al., ROS scavenging Mn₃O₄ nanozymes for in vivo anti-inflammation. *Chemical Science*, 2018. **9**(11): p. 2927-2933.
6. Lu, L., et al., Mn(3)O(4)nanozymes boost endogenous antioxidant metabolites in cucumber (*Cucumis sativus*) plant and enhance resistance to salinity stress. *Environmental Science-Nano*, 2020. **7**(6): p. 1692-1703.
7. Kresse, G. and J. Furthmuller, Efficient iterative schemes for ab initio total-energy calculations using a plane-wave basis set. *Physical Review B*, 1996. **54**(16): p. 11169-11186.
8. Perdew, J.P., K. Burke, and M. Ernzerhof, Generalized gradient approximation made simple. *Physical Review Letters*, 1996. **77**(18): p. 3865-3868.
9. Pisoschi, A.M. and A. Pop, The role of antioxidants in the chemistry of oxidative stress: A review. *Eur J Med Chem*, 2015. **97**: p. 55-74.
10. Conrad, M. and D.A. Pratt, The chemical basis of ferroptosis. *Nature Chemical Biology*, 2019. **15**(12): p. 1137-1147.
11. Singh, N., et al., A Redox Modulatory Mn₃O₄ Nanozyme with Multi-Enzyme Activity Provides Efficient Cytoprotection to Human Cells in a Parkinson's Disease Model. *Angewandte Chemie-International Edition*, 2017. **56**(45): p. 14267-14271.
12. Xue, Y., et al., Direct Evidence for Hydroxyl Radical Scavenging Activity of Cerium Oxide Nanoparticles. *Journal of Physical Chemistry C*, 2011. **115**(11): p. 4433-4438.
13. Fernandez-Garcia, S., et al., Enhanced Hydroxyl Radical Scavenging Activity by Doping Lanthanum in Ceria Nanocubes. *Journal of Physical Chemistry C*, 2016. **120**(3): p. 1891-1901.
14. Wu, H.H., et al., Hydroxyl radical scavenging by cerium oxide nanoparticles improves Arabidopsis salinity tolerance by enhancing leaf mesophyll potassium retention. *Environmental Science-Nano*, 2018. **5**(7): p. 1567-1583.
15. Schlick, S., et al., Scavenging of Hydroxyl Radicals by Ceria Nanoparticles: Effect of Particle Size and Concentration. *Journal of Physical Chemistry C*, 2016. **120**(12): p. 6885-6890.
16. Wu, H.H., N. Tito, and J.P. Giraldo, Anionic Cerium Oxide Nanoparticles Protect Plant Photosynthesis from Abiotic Stress by Scavenging Reactive Oxygen Species. *Acs Nano*, 2017. **11**(11): p. 11283-11297.
17. Zheng, K., et al., Regulation of ferroptosis by bioactive phytochemicals: Implications for medical nutritional therapy. *Pharmacol Res*, 2021. **168**: p. 105580.
18. von Kleist, L., et al., Role of the clathrin terminal domain in regulating coated pit dynamics revealed by small molecule inhibition. *Cell*, 2011. **146**(3): p. 471-84.
19. Robertson, M.J., et al., Synthesis of the Pitstop family of clathrin inhibitors. *Nat Protoc*, 2014. **9**(7): p. 1592-606.

REVIEWERS' COMMENTS

Reviewer #2 (Remarks to the Author):

My concerns have been properly addressed by the authors. The manuscript is now suitable for publication in Nature Communications in my opinion.

Reviewer #3 (Remarks to the Author):

The manuscript can be accepted.